# Hypertrophic cardiomyopathy disease results from disparate impairments of cardiac myosin function and auto-inhibition

Julien Robert-Paganin[1], Daniel Auguin [2] & Anne Houdusse [1]

Hypertrophic cardiomyopathies (HCM) result from distinct single-point mutations in sarcomeric proteins that lead to muscle hypercontractility. While different models account for a pathological increase in the power output, clear understanding of the molecular basis of dysfunction in HCM is the mandatory next step to improve current treatments. Here, we present an optimized quasi-atomic model of the sequestered state of cardiac myosin coupled to X-ray crystallography and in silico analysis of the mechanical compliance of the lever arm, allowing the systematic study of a large set of HCM mutations and the definition of different mutation classes based on their effects on lever arm compliance, sequestered state stability, and motor functions. The present work reconciles previous models and explains how distinct HCM mutations can have disparate effects on the motor mechano-chemical parameters and yet lead to the same disease. The framework presented here can guide future investigations aiming at finding HCM treatments.

[1] Structural Motility, Institut Curie, PSL Research University, CNRS, UMR 144, F-75005 Paris, France. [2] Laboratoire de Biologie des Ligneux et des Grandes Cultures (LBLGC), Université d'Orléans, INRA, USC1328, 45067 Orléans, France. These authors contributed equally: Julien Robert-Paganin, Daniel Auguin. Correspondence and requests for materials should be addressed to D.A. (email: auguin@univ-orleans.fr) or to A.H. (email: anne.houdusse@curie.fr)

ypertrophic cardiomyopathies (HCM) are the most prevalent inherited cardiac diseases, affecting one individual per 500[1]. Clinical manifestations range from asymptomatic to mechanical or electrical defects leading, in the most severe cases, to heart failure or sudden death[2]. Virtually all HCM is caused by mutation of genes necessary for cardiac muscle contraction and at least 80% of familial HCM is caused by mutation of two sarcomeric proteins: *MYH7* (ventricular β-cardiac myosin) and *MyBPC3* (cardiac myosin-binding protein C)[3]. More than 300 mutations in the β-cardiac myosin heavy chain gene are known to cause HCM in adults and mutations causing severe childhood HCM have also been described[1,4]. Although HCM mutations are found in all regions of β-cardiac myosin, clusters of mutations in regions such as the converter are known to be particularly severe[5]. The motor cycle of myosins couples ATP hydrolysis to actin-based force production upon the swing

of its distal region called lever arm, including the converter. The motor undergoes critical conformational changes and several structural states of different affinity for F-actin and different lever arm positions have been described[6] (Fig. 1a). To date, the effects of HCM mutations on motor activity and how they can lead to such a disease over the years have not been precisely deciphered[7]. While recent advances give hope in future treatments of such a disease with small molecule modulators of myosin function[8–12], it is essential to progress in understanding of the impact of β-cardiac myosin mutations to guide future therapeutic strategies.

Brenner and co-workers have been the first to propose a mechanism of HCM development[13–15] based on HCM mutations located in the converter, a motor subdomain critical for the swing of the myosin lever arm during force production[16]. They assessed that these mutations alter the mechano-chemical parameters of

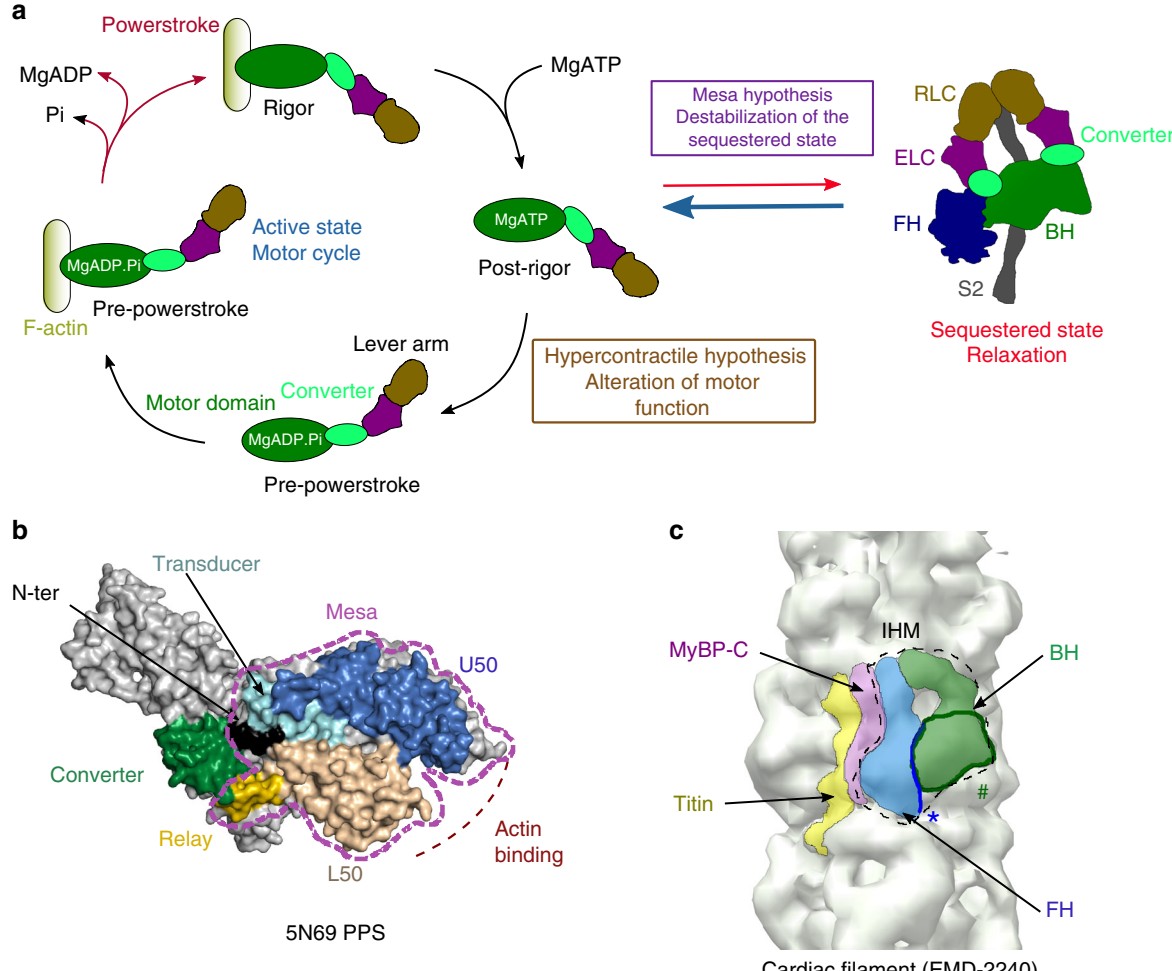

**Fig. 1** Myosin mesa and sequestered state. **a** Schematic representation of the motor cycle and the regulation of the β-cardiac myosin activity. On the left, when the motor detaches from the track upon ATP binding, the motor adopts the post-rigor (PR) state in which the lever arm is down and the motor has poor affinity for F-actin. During the recovery stroke, repriming of the lever arm leads to the pre-powerstroke (PPS) state in which hydrolysis can occur. The swing of the lever arm (powerstroke) upon reattachment of the motor to F-actin is coupled with the release of hydrolysis products. The nucleotide-free or rigor state has the highest affinity for F-actin. On the right, scheme of the sequestered state that is formed during relaxation. According to the mesa hypothesis, HCM mutations disrupt the sequestered state, increasing the number of myosin heads available to produce force. According to the Hypercontractile hypothesis, HCM mutations alter the myosin motor activity. **b** The mesa (purple dashed lines) is a long and flat surface of the myosin head composed of several myosin subdomains conserved in myosin IIs. **c** Electron density map of the human cardiac filament obtained from negative staining[23] (EMDB code EMD-2240). In the relaxed state, interactions between the blocked head (BH) and the free head (FH) of the myosin 2 dimer stabilize an asymmetric configuration. Extra densities of the filament correspond to other components of the thick filament, the cardiac myosin MyBP-C, and titin. Location of the mesa for each head indicates that the FH mesa (*) interacts with the BH while the BH mesa (#) is buried and interacts with components of the thick filament

individual myosin heads, thus modifying the power output of the cardiac muscle[13–15] (Fig. 1a). According to Brenner's results, HCM converter mutations alter the stiffness of the lever arm and thus modify the intrinsic force produced by each head, highlighting the essential role of the converter for motor compliance[14]. However, how a single point mutation in the converter can alter the global compliance of the molecular motor has yet to be described.

Recently, a contrasting hypothesis has been proposed by Spudich and co-workers who found that HCM mutations are strongly enriched on a flat surface of the myosin head, the mesa[17–19] (Fig. 1b). The mesa hypothesis[17,19] proposes that mutations of the mesa and converter regions trigger HCM by increasing the number of active heads since they lead to the destabilization of the myosin sequestered state, in which the heads are inactive and unable to interact with actin (Fig. 1a). In cardiac filaments, the sequestered state would be a way to regulate the number of active heads available to produce force during contraction[19]. In this state, the two heads of a myosin molecule form an asymmetric dimer (Fig. 1a) first described for the off state of smooth muscle myosin[20], which was later identified in myosins of invertebrate and vertebrate striated muscle, including ventricular cardiac muscle[21–24]. The widespread occurrence of the interacting heads motif (IHM) suggest a conserved mechanism to sequester these different myosins (Fig. 1c). In vertebrate muscles, such as cardiac, the sequestered state has been proposed to be the structural basis of a "super-relaxed" functional state that occurs during relaxation with extremely low ATPase activity[25–30]. The transition between the sequestered and the active states is driven by several parameters such as calcium concentration, interaction with MyBP-C, and mechanical load[31]. Current low-resolution structural models indicate that several HCM mutations could weaken the so-called IHM[19,32], thereby increasing the number of active heads. Recent data is consistent with this conclusion since the myosin intrinsic force is little affected by adult-onset HCM converter mutations[33] (Fig. 1a). Current models of the IHM lack sufficient resolution to describe the interfaces of the IHM at an atomic level[19] and how mutations can weaken or abolish the sequestered state. A higher-resolution structural model is thus required. In the context of HCM, the main challenge is to identify how different mutations can impair the cardiac myosin activity or its regulation and lead to the HCM disease.

Here, the cardiac myosin converter/essential light chain (ELC) interface is described for the first time from a 2.33 Å crystal structure of bovine β-cardiac myosin (98% identity with the human form). Insights on the role of the converter in myosin compliance and in stabilization of the sequestered state are provided from in silico studies of four severe HCM mutations located in the converter. In order to analyze how HCM mutations may destabilize the IHM or affect motor activity, an optimized molecular model of the sequestered state was built using our high-resolution cardiac myosin structures. Our results provide precise insights for the development of the HCM pathology, explaining how HCM mutations can modify motor activity and conformational plasticity of single heads and how they may also increase the number of heads available for force production by destabilizing the sequestered state.

## Results

**Structure of the β-cardiac myosin head**. We determined the 2.33 Å resolution structure of the Bovine Cardiac Myosin S1 fragment complexed with MgADP in the post-rigor (PR) state (PR-S1), a myosin conformation populated upon ATP-induced detachment of the motor from actin (Fig. 2a, Table 1). The same fragment was previously crystallized in the pre-powerstroke (PPS)

state complexed with omecamtiv mecarbil (OM)[11] (OM-PPS-S1, PDB code: 5N69) (Supplementary Fig. 2A). The ELC bound to the IQ motif was built ab initio from well-defined electron density (Supplementary Fig. 1A).

These structures thus allow the description of the critical ELC/converter interface for two different positions of the cardiac myosin lever arm. There is no major variation in the lever arm coordinates of the PR and the PPS structures (RMSD 0.98 Å) (Supplementary Fig. 2B). The pliant region (R777-783) at the junction between the converter and the IQ motif/ELC[16] is straight in both structures (Supplementary Fig. 2B). In the two structures, interactions between the ELC and the converter are mainly electrostatic (Fig. 2b, Supplementary Fig. 2A) via a network of charged residues that contribute to the stabilization of the lever arm.

The overall fold of the converter is similar to that found for smooth and invertebrate striated muscle myosins, with the exception of two loops displaying sequence variations, that we define as the top loop and the side loop (Fig. 2c–e). The side loop appears to assist stabilization of the fold of the converter, while the top loop forms a part of the converter/ELC interface. Thus, these structures provide complete atomic coordinates that are essential to evaluate the structural plasticity of the lever arm and to study the consequences of specific missense mutations in this region.

**The cardiac lever arm internal dynamics**. The dynamics within the cardiac myosin lever arm were studied in 30 ns molecular dynamics experiments (see Methods) using coordinates from the PR-S1 lever arm structure corresponding to the converter (701–777) and 1st IQ motif (778–806) as well as the bound ELC. We computed the root-mean-square deviation (RMSD) fluctuation during the simulation for all Cα atoms and represented it with the putty representation[34] (Fig. 3). The WT simulation reveals notable internal dynamics of the converter (RMSD of Cα between 0.6–5.8 Å) linked to large fluctuations in the top and side loops (Fig. 3a). The converter/ELC interface does not behave as a stable supramodule. In fact, dynamic fluctuations promote formation of labile polar bonds depending on the relative orientation of the ELC and converter. Some residues increase the dynamics at the converter/ELC interface by oscillating between partners, and thus favoring rearrangement. For example, the top loop residue $_C$E732 oscillates between $_C$R723 and $_{ELC}$R142 by a mechanism analogous to musical chairs: a single negative charge ($_C$E732) alternates between two nearby positive charges ($_C$R723 and $_{ELC}$R142) (Fig. 3a). These labile bonds regulate the conformation and dynamics of the top loop yielding controlled dynamics at the converter/ELC interface. In addition, the space explored by the top loop is also controlled by the side chain of $_C$R719 interacting with carbonyls of the top loop in certain conformations that are labile due to the top loop dynamics. In contrast, the stable interaction of $_P$R780 with the ELC ($_{ELC}$D136), constrains the dynamics of the pliant region[16], controlling the myosin lever arm deformability. To further explore the interactions important to control the lever arm structural plasticity, we have chosen the study in silico the R719G and R780E mutations. These mutants have not been reported as associated with HCM but we had anticipated that they could act as anchoring residues limiting the dynamics of the lever arm. Both of these test mutations indeed result in drastic increase in the amplitude of the converter dynamics via destabilization of contacts at the pliant region (Supplementary Fig. 3A, B, C). This study thus highlights that the dynamics in this multi-domain lever arm is controlled by crucial bonds within the lever arm. However, the lever arm is dynamic rather than rigid, and modulation of the conformations of the top

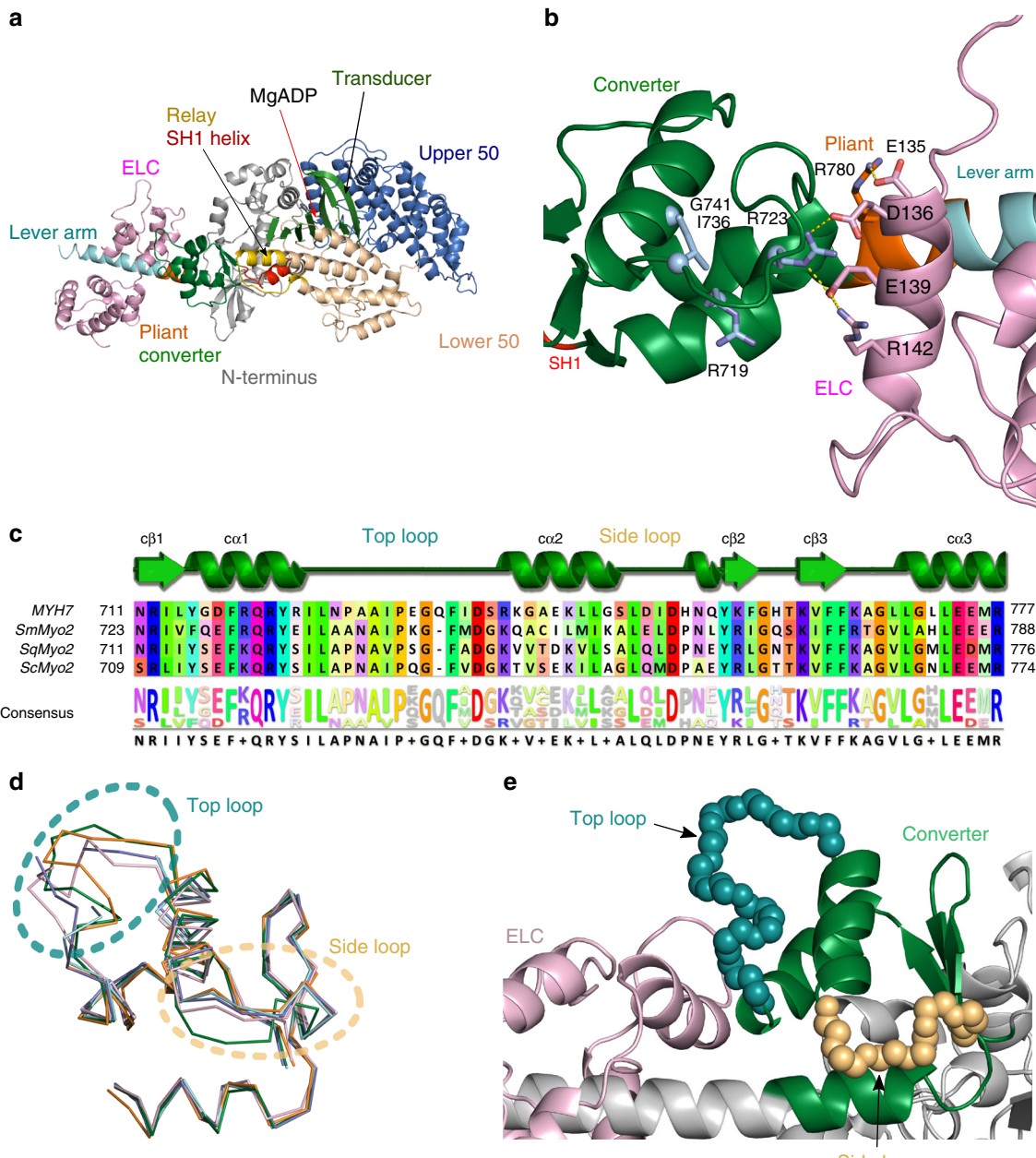

**Fig. 2** Crystal structures of β-cardiac myosin and description of the converter/ELC interface. **a** X-ray structure of β-cardiac myosin S1 complexed with MgADP in the post-rigor state (PR-S1). **b** Interface between the converter and the ELC as found in the PR state. This interface involves mainly electrostatic interactions between a cluster of negatively charged residues of the ELC (D136; E135 and E139) and positively charged residues of the heavy chain (R723 and R780). Side chains of interacting residues (sticks) and polar interactions (yellow lines) are represented. Four converter mutations studied in this work (R719W; R723G; I736T; G741R) are colored in light blue. **c** Structure alignment of converters of the Myosin 2 superfamily. MYH7: bovine (*Bos taurus*) β-cardiac myosin; ScMyo2: bay scallop (*Argopecten irradians*) Myosin 2; SqMyo2: longfin inshore squid (*Doryteuthis pealeii*) Myosin 2; SmMyo2: chicken (*Gallus gallus*) gizzard smooth muscle myosin 2. **d** Superimposition of several Myo2 converter structures: from PR-S1 colored in green; from SqMyo2 (PDB code 3I5F, pink); from SmMyo2 (PDB code 1BR1, orange); from ScMyo2 in the PPS (PDB code 1QVI, purple) and Rigor (PDB code 1SR6, cyan) states. **e** Location of the top loop (deep teal blue) and the side loop (sand yellow) in the cartoon representation of the crystal structure of the β-cardiac myosin PR-S1 (gray) with the converter colored in light green and the ELC colored in light pink. The top loop is part of the converter/ELC interface

loop of the converter as well as of the pliant region greatly contribute to the dynamics at the converter/ELC interface.

**Converter HCM mutations alter the lever arm dynamics**. We next searched with simulations how the controlled dynamics observed in the WT lever arm are affected by severe HCM converter mutations. We focused on three converter mutations for

which single-molecule studies had been performed. The intrinsic force was virtually unchanged by the G741R mutation and was only modestly (15–30%) decreased for R719W and R723G[33] (Supplementary Fig. 4). An additional effect of these mutants on myosin regulation cannot be excluded since some HCM mutations have already been reported to disrupt interfaces of the IHM[35]. In silico studies of these mutants show a large effect on

**Table 1 Data collection and refinement statistics**

| | PR-S1 |
|---|---|
| *Data collection* | |
| Space group | P1 |
| Cell dimensions | |
| *a, b, c* (Å) | 55.12, 94.32, 125.41 |
| *α, β, γ* (°) | 104.76, 92.34, 100.06 |
| Resolution (Å) | 47.97–2.33 (2.413–2.33)[a] |
| $R_{meas}$ | 0.16 (1.35) |
| $I/\sigma I$ | 7.76 (1.09) |
| $CC_{1/2}$ (%) | 99.4 (43.3) |
| Completeness (%) | 98.46 (97.87) |
| Redundancy | 3.6 (3.7) |
| *Refinement* | |
| Resolution (Å) | 47.97–2.33 |
| No. of reflections | 359,796 (total), 100,862 (unique) |
| $R_{work}/R_{free}$ (%) | 18.95/22.69 |
| No. of atoms | |
| Protein | 14,655 |
| Ligand/ion | 79 |
| Water | 74 |
| *B*-factors | |
| Protein | 68.85 |
| Ligand/ion | 38.72 |
| Water | 47.82 |
| R.m.s. deviations | |
| Bond lengths (Å) | 0.015 |
| Bond angles (°) | 1.79 |

[a]Values in parentheses are for highest-resolution shell

the top loop conformation and a reduction of the lever arm pliancy.

The R719 side chain interacts transiently with the top loop and helps in maintaining its conformation (Fig. 3a). The R719W mutation has two effects, first there is a destabilization of the top loop conformation found in WT, and, second, the contacts promoted by the bulky W719 decrease the internal dynamics of the converter and of the converter/ELC interface (Fig. 3b). The R723 side chain is part of the network of labile converter/ELC interactions that facilitate dynamics in the WT lever arm (Fig. 3a). The R723G mutation removes a charged residue crucial for this interface as well as for the stabilization of the top loop conformation. Interestingly, in this mutant, the ELC/converter interface is defined via formation of hydrophobic interactions and the top loop is not part of this interface. While the internal dynamics of the R723G converter is locally high due to destabilization of the top loop, drastic loss of the controlled "musical chairs" dynamics at the converter/ELC interface leads to more stiffness overall (Fig. 3c). The G741R mutant introduces a charged and bulky side chain in the second helix of the converter which results in an extra charged partner at the converter/ELC interface that disrupts the musical chairs dynamics found in WT (Fig. 3d). Thus, the top loop $_C$E732 residue can now interact with this mutant R741 side chain and is no longer confined to oscillations between $_C$R723 and $_{ELC}$R142. This leads to a shift in the position of the top loop that reduces the internal dynamics of the G741R converter and the ELC/converter interface compared to WT (Fig. 3d).

Overall, these simulations show that although the lever arms of these HCM mutant myosins are slightly stiffer compared to WT in terms of structural plasticity, the effects of these HCM causing mutations are different compared to test mutations (R719G and R780E) which resulted in a drastic increase of the structural plasticity of the lever arm. The modest effects of the three HCM

mutations on mechano-chemical parameters of S1 fragments[33] confirm that the resulting slight decrease in lever arm compliance keeps the motor functional. The main effect of these three mutations is on the top loop conformation and its dynamics, which result in disruption of the dynamic network at the converter/ELC interface. We also studied a fourth HCM mutant[36] that affects a top loop residue, I736T[34] (Supplementary Fig. 3D). Only local changes in the top loop conformation result from this mutation leading to limited consequences on the dynamics of the converter or the interface with the ELC. Since, of the four pathological mutations investigated, the effect on compliance of the lever arm is modest and the changes in dynamics are focused on the top loop, we further our analysis by investigating how HCM mutations affect the formation of the β-cardiac myosin sequestered state.

**Sequestered state model.** In the sequestered state, myosin heads are in a conformation with the lever arm primed, which is a hallmark of the PPS state[20,24]. Therefore, the cardiac PPS structures we have determined provide the best starting model to date to generate a high-resolution in silico model of the cardiac myosin sequestered state (see Methods). By a combination of homology modeling, ambiguous docking methodology[37], molecular dynamics, and fit in the two existing maps[23] (see Methods), we have built a model of the bovine β-cardiac myosin sequestered state (Supplementary Data 1) that can provide a much improved description of the inter-head interactions responsible of its stabilization (Fig. 4a). Since the human and the bovine β-cardiac myosins are very close in sequence identity, we also humanized the bovine model (see Methods, Supplementary Data 2). The polymorphism between these two models does not lead to any difference for the interfaces that stabilize the IHM (see Supplementary Fig. 6). The model perfectly fits in current low-resolution available maps (Supplementary Fig. 5). This model is the best to date for two reasons (see Methods for a detailed comparison with previous models): (i) it is based on a high-resolution PPS state structure[11] (PDB code 5N69) that is precise enough to ensure the correct position of motor domain residues; (ii) the use of molecular dynamics allowed us to model realistic inter-head interactions and to ensure that the our final model displays good geometrical parameters.

In the quasi-atomic IHM model, consistent with previous models[32,35] (PDB code 5TBY; MS01), the sequestered state is asymmetrical and composed of two folded heads (Fig. 4a). The S2 coiled-coil interacts with the mesa of the so-called blocked head (BH) (Fig. 4a, c). The BH is unable to interact with actin since a part of its actin-binding interface is docked on the free head (FH) via the flat surface mainly composed of the FH-mesa and FH-converter (Fig. 4b, d). We mapped on the surface of the mesa and the converter of the FH and the BH the regions that are involved with inter-molecular or intra-molecular interactions (Fig. 4c, d). The FH-converter interacts with the BH head, while the BH-converter putatively interacts with MyBP-C. The BH/MyBP-C interface is supported by previous experiments[35] and the architecture of the cardiac filament[23]. There are two main differences in the sequestered model we propose compared to previous models: (i) the motor domain coordinates are much improved and reliable since the molecular dynamics have optimized the overall geometry of the model and the intra-molecular interactions that occur upon head/head recognition; (ii) the lever arm differs drastically in the pliant region for both heads. In previous models, the use of the smooth myosin structure (1BR1) as a template had imposed a bend at the pliant region (Supplementary Fig. 6A), which is not observed in the cardiac myosin PPS X-ray structure.

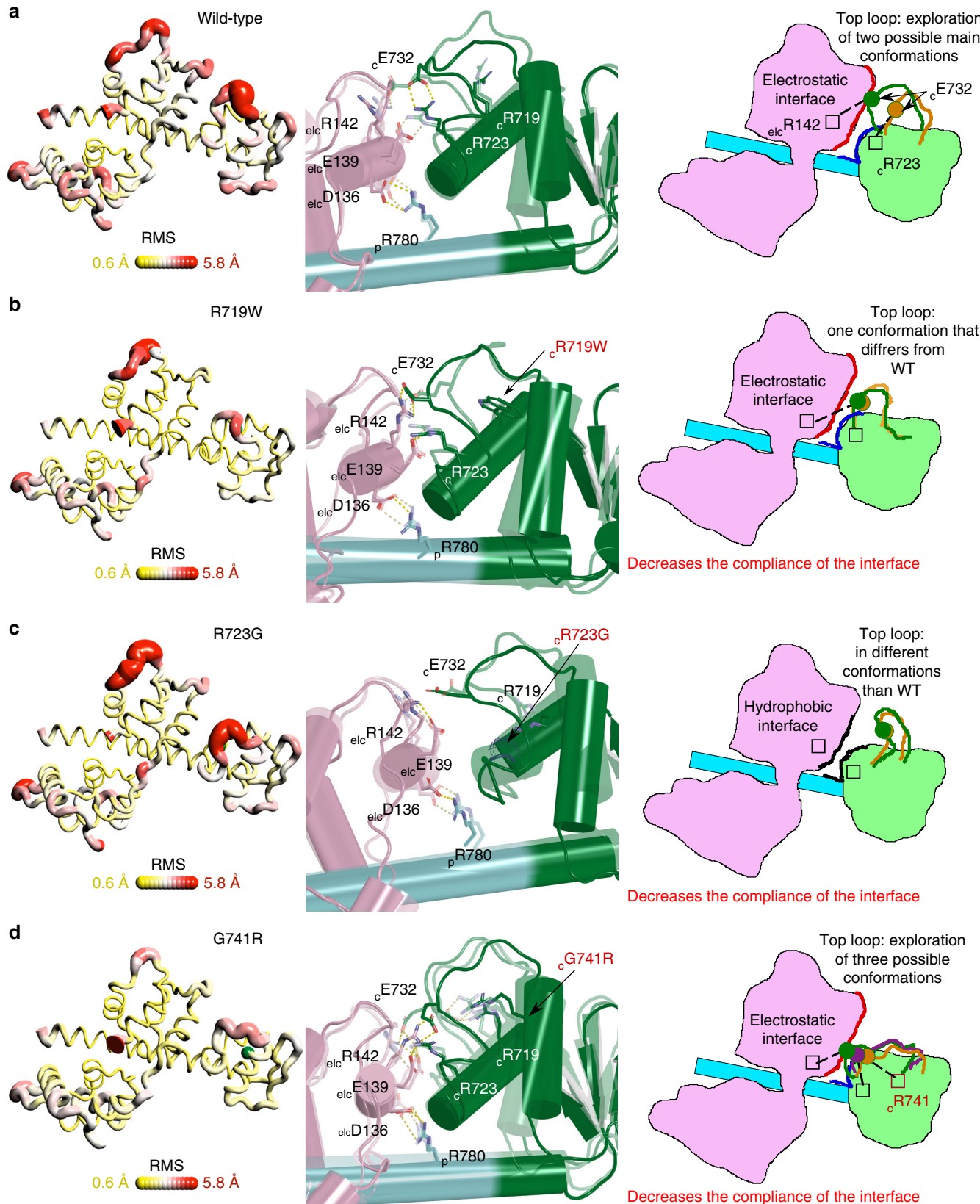

This model provides atomic-level description of inter-head interactions in the sequestered state. A remarkable feature of the FH/BH interface is the major role of the FH-converter and specifically the top loop: (i) I736 (FH top loop) occupies a hydrophobic pocket of the BH-U50 in a lock-and-key fashion (Fig. 4b, Supplementary Fig. 6E), (ii) polar and charged residues from the FH-converter, including R719 (converter) and R739 (top loop), create an extended network with the BH-U50 (Fig. 4b).

**Fig. 3** Dynamics of the converter/ELC interface. Schematic representation of the results from the molecular dynamics simulations for the wild-type (WT) and the three HCM mutants that have been analyzed in silico: R719W, R723G, G741R. On the left, a "putty representation" of the β-cardiac myosin converter and first IQ (aa 701–806) bound to the ELC. RMS fluctuations during 30 ns simulations are represented with RMS. Scale ranging from 0.6 (in yellow) to 5.8 Å (in red). On the center, the structure of the interface between the ELC, the converter, and the pliant region is represented, as well as the position of key residues maintaining the interface and its plasticity. In each structure, the position of the residue mutated is labeled in red. On the right, a schematic representation of the region containing the converter, the ELC, and the lever arm is displayed. The different populations of the top loop allowed by the dynamics of this region are drawn and the nature of the interactions between the converter and the ELC is also represented. In each case a state is represented opaque and the others are in transparency in order to best compare the different populations (positions of the top loop are colored differently). On the center and on the left, the myosin subdomains are colored differently: the converter in green; the IQ region in cyan; and the ELC in light pink. **a** WT, **b** R719W, **c** R723G, **d** G741R

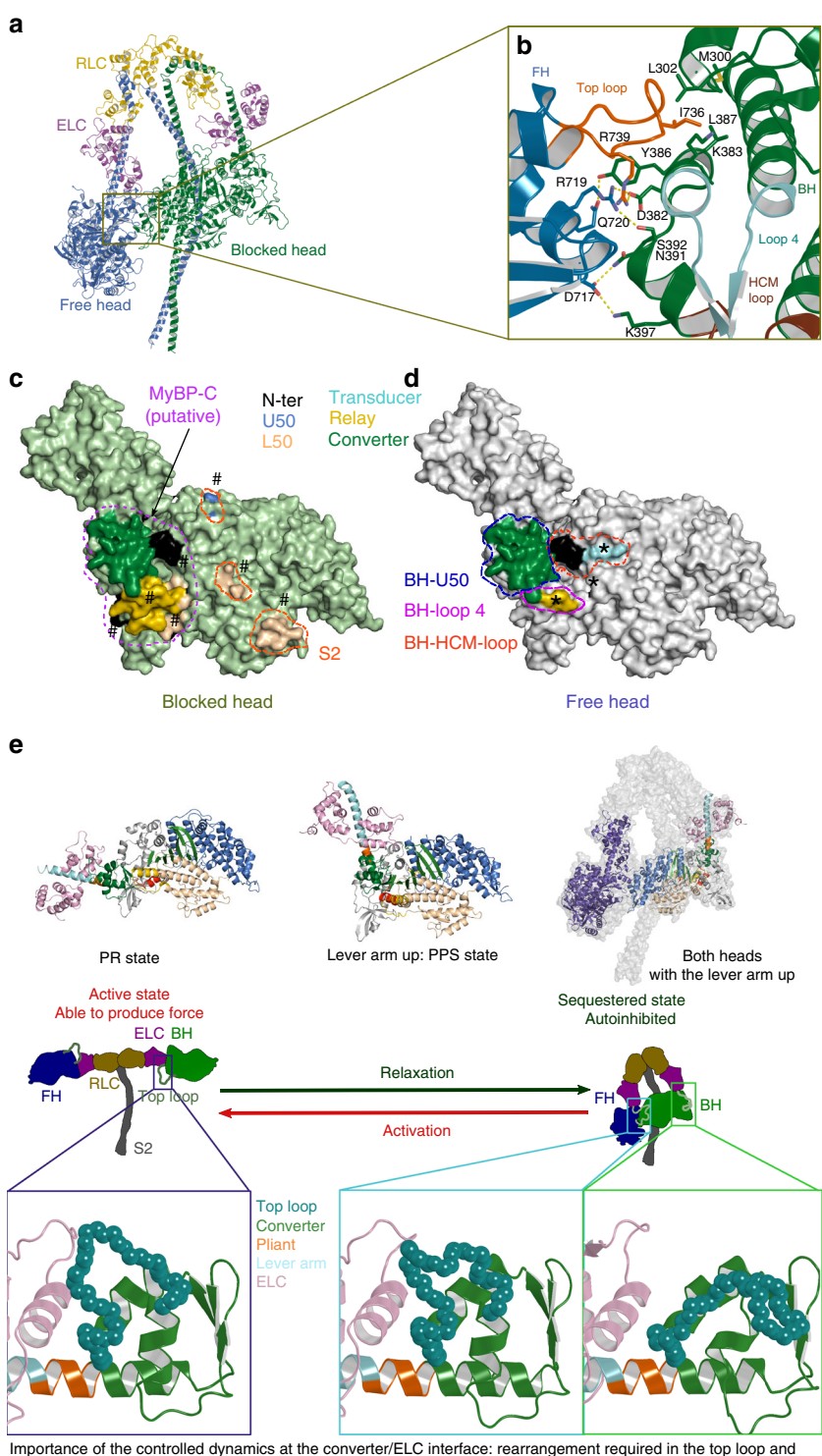

Importance of the controlled dynamics at the converter/ELC interface: rearrangement required in the top loop and the converter/ELC to promote two differences interfaces for the BH and the FH heads

**Fig. 4** An optimized model of the sequestered state of β-cardiac myosin. **a** β-cardiac myosin sequestered state modeled from the cardiac myosin S1 PPS structure. This model results from optimization of the intra-head interactions that occur upon formation of the IHM[59] with two asymmetric heads: the free head (FH) and the blocked head (BH). **b** Detailed analysis of a region of the interface involving the FH-converter and the BH-U50 subdomain. Note that the top loop plays a major role. **c** Surface of the FH-mesa and the FH-converter involved in interactions with the BH. **d** Surface of the BH-mesa and the BH-converter involved in interactions with the FH. A putative surface of interaction with the partner MyBP-C is also represented. The regions of the interface that are part of the mesa are labeled with a * and a # on the FH and on the BH, respectively. **e** Schematic representation analyzing the prerequisites to form the IHM. On top, different myosin conformations illustrate the differences in position of the lever arm in the PR and the PPS states. Two heads in the PPS conformation are shown in the sequestered state. On the center, a schematic representation of the two states adopted by the heavy meromyosin (HMM) is shown. On the bottom, the different conformations of the top loop present in the PR crystal structure and in the BH and the FH of the sequestered state are displayed in cartoon representation. To form the sequestered state, the lever arms of the FH and BH heads must adopt conformations, in particular for the top loop. Note, however, that the pliant region remains close to the conformation of the WT PPS structure. Dynamics at the ELC/converter interface of the two heads of the myosin dimer is thus important to promote sequestration of the heads

These interactions significantly differ from the previous models in which the top loop is in a drastically different conformation and was not predicted to form contacts that would mobilize I736 (Supplementary Fig. 6E).

Finally, this model highlights a role in the lever arm dynamics for the formation of the sequestered state (Fig. 4e). In relaxing conditions, formation of the interactions between heads requires that both heads adopt an asymmetric conformation, in particular in the hinges of their lever arm, while the two motor domains position the lever arm up (PPS state). Interestingly, the converter/ELC interface and in particular the conformation of the top loop is different in the BH and the FH heads. This implies that controlled dynamics at the converter/ELC interface is important for the establishment of the inter-head interfaces and thus mutations affecting the dynamics can impair the formation and/or the stability of the sequestered myosin state.

**Implication of the cardiac IHM model in disease**. The atomic model of the sequestered state built from high-resolution cardiac structures and optimization of the intra-molecular contacts provides a powerful structural framework to investigate the impact of HCM mutations. We have carefully studied a set of 178 mutations previously described as triggering the HCM phenotype (www.expasy.ch,[18,32]; see Supplementary Data 3). We evaluated the effects of each of these mutations based on the structural myosin models that define how myosin can produce force[6] and how it is sequestered for inactivation (Supplementary Table 1, Supplementary Data 3). This analysis resulted in a classification of the mutations in six classes that provide an overview of the different molecular consequences of HCM-causing mutations that can trigger the disease (Fig. 5a, b). Each proportion evoked in this section is calculated as a percentage of the entire set of 178 selected mutations.

As in the mesa hypothesis, some mutations are predicted to affect the IHM stability since they impact residues found at the interfaces that stabilize the off state[32,35] (Supplementary Table 1, Fig. 5a, b). Among these mutations, 18% of the set represent mutations that destabilize the IHM with no significant effect on the motor function (Fig. 5a). Some mutations that directly affect the IHM stability also have effect on motor function (Fig. 5a, 16.5% of the set). These mutations are located in the transducer, the actin-binding interface or the converter which are regions of interest in the mechano-chemical cycle of the motor (Supplementary Table 1). Interestingly, a large number of mutations affecting the sequestered state (31% of the set) are not directly located at the interfaces of the IHM but are predicted to alter the stability of the PPS conformation that is necessary to form the IHM (Fig. 5a, Supplementary Table 1). In this case, the effects of the mutations are diverse, they can affect regions important for allosteric communication during the motor cycle (Switch I, Switch II, relay or SH1 helix), destabilize interactions between

regions necessary to maintain the PPS state, or alter the compliance of the lever arm at the pliant region (Fig. 4e) required for the IHM formation.

In addition, this analysis defines three other classes of mutations that impact the motor function and/or its structural stability (34.5% of the set). Among these, an important class regroups mutations with no effect on the IHM stability while strong effects on motor function are predicted (Fig. 5a). This class that regroups 7.5% of the set is particularly interesting since it predicts that alteration in motor function rather than on the number of heads available for contraction can also lead to HCM. One of the mutations, A426T, occurs at a position that has been described in Myosin VI (A422L) to slow the transition after Pi release[38]. This mutation indeed introduces a bulky residue in the 50 kDa cleft that must close during the powerstroke to bind actin strongly. This indicates that some of these mutations can lead to HCM by altering purely the states and transitions populated during the powerstroke. Another set of mutations is predicted to alter significantly both motor function and its stability (Supplementary Table 1, Fig. 5a, 18.5% of the set). Since mutations from this class are predicted to affect the stability of the motor, we cannot exclude that some of these mutations may result in decrease stability of the PPS and consequently on the IHM. Finally, we defined a last class (Supplementary Table 1, Fig. 5a) with 8.5% of the mutations for which only mild alteration of the local structure or the function of the protein is predicted.

Altogether, our results show that HCM mutations can have a wide variety of effects. Indeed, these mutations are found in all the subdomains of the myosin heavy chain (Fig. 5b, c, Supplementary Data 3) and interestingly mutations close in space can belong to distinct classes and induce very different effects. This illustrates the complexity of the HCM pathology and the fact that HCM mutations can have not only effects on the IHM but also on motor function. While awaiting further biophysical measurements (currently limited to a small set of mutations[33,39–42]; Supplementary Data 3), the classification of these mutations using structural models will open avenues to investigate how altered motor function and/or its regulation can lead to HCM.

## Discussion

Molecular dynamics allowed us to describe the key role of the top loop and of electrostatic interactions in maintaining and controlling the converter/ELC interface dynamics. Here we highlight that loss of this controlled dynamics can occur from multiple single mutations at the converter/ELC interface or at the pliant region, which corresponds to the highly deformable hinge point between the converter and the IQ/ELC modules of the lever arm[16]. In this regard, dynamics in the lever arm follow from all interactions that either regulate the converter fold stability or the

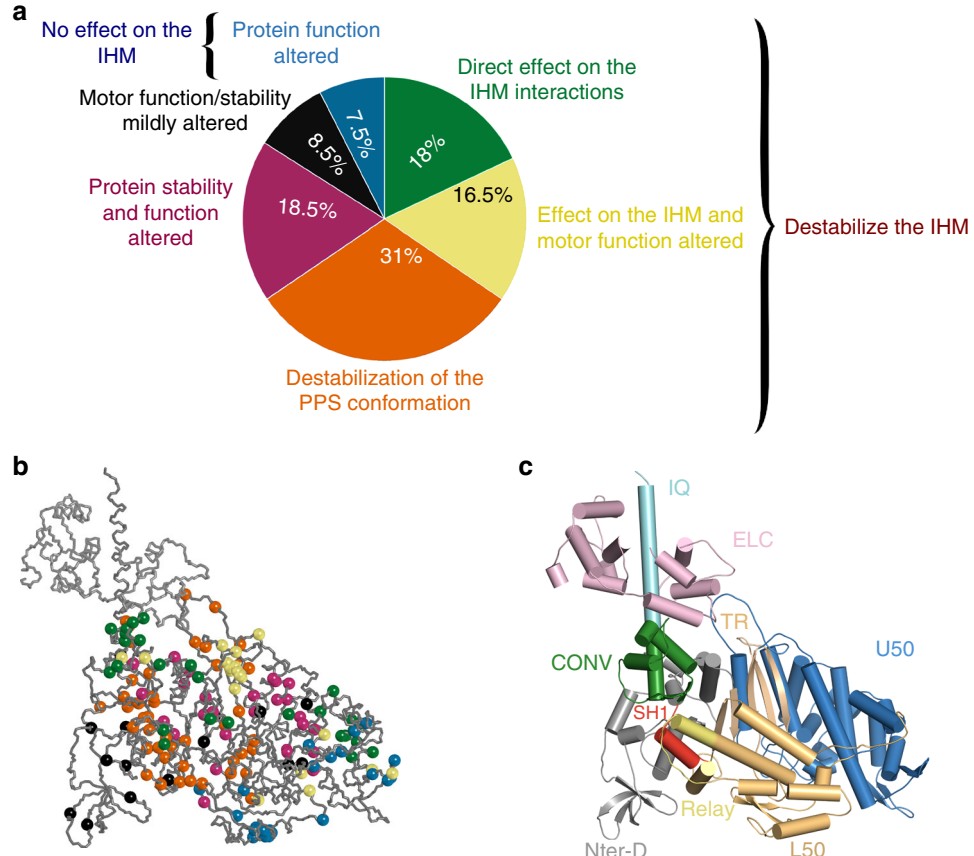

**Fig. 5** Structural and functional consequences of 178 HCM mutations. **a** The chart pie represents the proportion of mutations belonging to six classes depending on their effect on the structure, the function and the stability of the IHM. These six classes are: mutations destabilizing the IHM (green); mutations destabilizing the IHM and the motor function (yellow); mutations destabilizing the PPS conformation and the IHM (orange); mutations that alter the protein function (blue); mutations that alter the protein function and the protein stability (purple); and mutations that mildly affect protein function or stability (black). **b** Positions of the HCM mutations on the PPS structure of the β-cardiac myosin (PDB code: 5N69). Mutations are represented as balls colored depending on their predicted consequences, following the color code defined in **a**. **c** Cartoon representation of the PPS structure of the β-cardiac myosin with all the subdomains colored. TR transducer, ELC essential light chain

interactions that form between the converter and IQ/ELC modules of the lever arm.

The in silico study of a set of converter mutations highlights the previously unanticipated controlled dynamics that naturally occur between the converter and the ELC which must be sufficiently rigid for myosin activity but also sufficiently pliant to allow the formation of the sequestered state of β-cardiac myosin. Thus, this study not only supports the previous assumption that the converter is essential in maintaining the proper stiffness of the lever arm[13]; it indicates how the inherent complex interplay between residues of the converter/ELC can be easily dysregulated. Interestingly, a different point mutation at a particular position (R719W and R719G) can have antagonistic effects on the lever arm pliancy. This illustrates how the sequence of this region finely tunes the lever arm dynamics and validates the in silico approach to investigate the effect of mutations on the dynamics of this region. The converter HCM mutants that we analyzed have two effects (i) they modify the conformation of the top loop and (ii) they slightly increase (R719W, R723G, G741R) or have no effect (I736T) on the stiffness of the lever arm. This agrees with previous single-molecule measurements that determined that R719W, R723G, and G741R have only modest effects on force production of S1 fragments[33] (less than 30% decrease) (Supplementary Fig. 4).

The optimized quasi-atomic sequestered model (Fig. 4a) we built from atomic structures of the cardiac head allows us to predict the consequences of a set of 178 HCM mutations and demonstrates that this disease may occur from either dysregulation of the motor activity itself or the dysregulation of the sequestered state that shuts down myosin activity. Our results support the previous hypothesis that HCM mutations can disrupt the IHM and thus increase the number of heads available to participate in force production[17]. This is the case for at least 65.5% of the set we examined (Fig. 5a). While some HCM mutations can directly affect the contacts within the IHM, other mutations are predicted to affect the IHM stability (Supplementary Data 3), in particular by altering the stability of the PPS state of the myosin heads, which is the structural state adopted by both heads in the sequestered state (Fig. 4e). Several mutations can have dual effects; it is the case, for example, of the extensively studied R403Q located in the so-called HCM-loop involved in actin binding. Single-molecule studies report that the R403Q mutant has a decreased affinity for actin in the strongly bound states and displays mechanical defects[42]. While the motor function is affected[42], this mutation likely also weakens the IHM since the HCM loop of the BH interacts with the N-ter subdomain and transducer of the FH (Fig. 4d). In addition, our results clearly explain the consequences of the selected set of HCM converter

mutations studied in molecular dynamics. These mutations destabilize the sequestered state by three mechanisms (i) they all affect the dynamics and the conformation of the top loop which is a part of the inter-head interface; (ii) at least two of them (I736T and R719G) change a residue directly involved in the interface; and (iii) three of these mutations (R719W, R723G, and G741R) decrease the dynamics of the converter/ELC interface which is necessary to form the sequestered state since the conformation of this interface is different in the FH and in the BH compared to the active state (Fig. 4e). Then, according to our results, even if the R719W and R723G mutations induce a small decrease in force production of individual heads[31], they also destabilize the sequestered state and increase the number of heads available to produce force, bringing a possible explanation for why fibers from patients carrying these mutations produce more force[13–15].

An unanticipated result from this study is the description of an interesting class of mutations (7.5% of the set) that effects motor activity without any predicted effect on the IHM stability. Interestingly, previous work proposed that HCM mutants can be differentiated from other cardiac pathologies since they would mainly disrupt the IHM while dilated cardiomyopathies (DCM) mutations would mainly alter myosin function, often being located close to the active site[32]. According to our results, this statement has to be nuanced since 7.5% of the HCM mutations we have studied are also predicted to only alter myosin function. In addition, some HCM mutations in this class involve positions close to the active site (for example, Y115H and T124L). DCM mutations are predicted to have a strong effect in diminishing the power output of cardiac contraction (Supplementary Data 4). Future studies on cardiac myosin under load[12] are essential. Important future functional studies are required to define more precisely the frontier between HCM and DCM at a molecular scale. In particular, it will be critical to test whether the HCM mutations located close to the active site, in the 50 kDa cleft or at the actin-binding interface (Fig. 5a) result in gain of power output of cardiac contraction, in contrast to DCM mutations which are predicted to result from a deficit in motor function[32,43] (Supplementary Data 4). Studies of atypical mutations that lead to different heart impairment will also be of particular interest to investigate how the fate of cardiac cells depend on specific loss of myosin function. This is the case for the R243H mutation which leads to an apical HCM and DCM disease, or of three DCM mutations which are predicted to directly destabilize the IHM (Supplementary Data 4).

Recently, the discovery of small molecules able to modulate the activity of β-cardiac myosin has provided new pharmaceutical perspectives for treatment of cardiac diseases[8–11]. The mechanism of action of a cardiac myosin activator, OM has been elucidated: OM favors the PPS conformation and thus increases the number of heads able to participate in force production upon the systole on-set[11]. Interestingly, a recent study has concluded that OM is not compatible with the sequestered state of cardiac myosin while the myosin inhibitor blebbistatin (BS) favors the sequestered state[44]. Our results are consistent with this study since the site of BS is internal within the motor domain and is not affected by IHM contacts. Thus, BS binding is compatible with the sequestered state. The fact that BS favors the off state[44] is a strong argument in favor of the model previously proposed[45] from low-resolution EM maps which we have defined at a better resolution here with both heads mainly unaffected for the motor domain but which uses hinges at the converter/ELC interface to allow the asymmetric association of the myosin heads to form the IHM (Fig. 6). In contrast, the binding site of OM is located in a pocket between the converter and the N-terminal subdomain[11] and OM binding restricts the plasticity of the converter, leading to a decreased ability of the heads to adopt the sequestered state

(Fig. 6). Thus, stabilization of the PPS of the motor is not sufficient to favor the myosin sequestered state and shutting down of motor activity during relaxation. Discovery of drugs that directly affect or favor the formation of the IHM motif would be of exquisite use to control more precisely the motors available. The inhibitor Mavacamten, which has been reported to favor the sequestered state conformation of β-cardiac myosin, would be a good candidate for such a treatment[29,30].

In this work, we indicate that HCM mutations can have different effects on the motor activity and on the regulation of cardiac myosin. It is thus unlikely that a unique drug can be used to treat all classes of HCM. Critical functional studies are required to identify how different mutations lead to impairment and how drugs modifying myosin activity or the stability of the PPS state may correct the effect of genetic mutations that belong to different classes. Since some HCM mutations may also impair the binding site of myosin drug modulators, it is critical to investigate how diverse specific drugs can restore myosin activity upon binding to diverse pockets of the myosin motor. The current knowledge about the rearrangements within the myosin motor upon force production[6], the development of precise methodologies to study these mutations[12,33,46], and the first in vivo studies about how drug may slow the progression of HCM disease[10] announce that research will bring new therapies for this disease in the near future.

## Methods

**Protein purification.** Bovine cardiac fragment S1 has been purified from fresh heart (Pel-Freez Biologicals) via the method described by Planelles-Herrero et al[11]. Subfragment-1 was prepared from full length by limited chymotryptic digestion in the buffer (20 mM K-Pipes, 10 mM K-EDTA, 1 mM DTT, pH 6.8). The solution containing the protease (tosyl-Lysyl-chloromethane hydrochloride (TLCK)-treated α-chymotrypsin; Sigma) was incubated at 22 °C for 30 min, and the digestion was stopped by the addition of 1 mM phenylmethylsulfonyl fluoride (PMSF). Insoluble myosin rods were removed by centrifugation (29,000 × $g$, 30 min, 4 °C). Cardiac S1 fragment was precipitated using ammonium sulfate (60% w/v final) and centrifugation (29,000 × $g$, 30 min, 4 °C). The pellet was resuspended and dialyzed against low-salt buffer (12 mM K-Pipes, 2 mM MgCl₂, 1 mM DTT, 0.1 mM PMSF, pH 6.8). The S1 was further purified by anion-exchange chromatography on Mono-Q (GE Healthcare) at 4 °C in the buffer 20 mM Tris-HCl, 0.8 mM NaN₃, pH 8 using a 0–350 mM gradient of NaCl. Fractions containing the purified S1 were pooled and buffer-exchanged into the buffer 10 mM HEPES, 50 mM NaCl, 1 mM NaN₃, 2.5 mM MgCl₂, 0.2 mM ATP, 1 mM TCEP, pH 7.5. The final S1 was concentrated up to 20–30 mg ml⁻¹, supplemented with 2 mM MgADP, aliquoted and flash-frozen in liquid nitrogen before storage at −80 °C.

**Crystallization and data processing.** Bovine cardiac fragment S1 has been purified from fresh heart as described in ref. [11]. Crystals of bovine cardiac fragment S1 (25 mg ml⁻¹) were obtained at 4 °C by the hanging drop vapor diffusion method from a 1:1 mixture of protein, 2 mM MgADP and precipitant containing 12.5% PEG mix MMW; 10% glycerol; 7.5% Sodium-Tacsimate, pH 6.0; 3.3% DMSO; 0.5 mM TCEP. PEG mix MMW was prepared by mixing stock solutions (50%) of PEG 2 K, 3350, 4 K and 5 K MME in equal volume[47]. Crystals were transferred in the mother liquor containing 25% glycerol before flash freezing in liquid nitrogen. X-ray diffraction data were collected at 100 K on the ID23.2 beamline ($\lambda$ = 0.87290 Å) at the ESRF synchrotron (Grenoble). Diffraction data were processed using XDS package[48]. The crystals belong to the P1 space group with two molecules per asymmetric unit. The cell parameters and data collection statistics are reported in Table 1.

**Structure determination and refinement.** Molecular replacement was performed with human cardiac myosin (residues 3–706) without water and ligand (PDB 4P7H) with Phaser[49] from the CCP4i program suite. Manual model building was achieved using Coot[50]. Refinement was performed with Buster[51]. The statistics for most favored, allowed, and outlier Ramachandran angles are 96.97, 2.81, and 0.22%, respectively. Structure determination indicated that this crystal form corresponds to a PR state. The final model has been deposited on the PDB (PDB code 6FSA).

**Human β-cardiac myosin.** In this work, we chose to use the bovine β-cardiac myosin. This choice has been done because the S1 fragment crystal structures to date are from bovine β-cardiac myosin (PPS, PDB code 5N69, and PR, PDB code 6FSA). Human and bovine β-cardiac myosins are very close and share 98% identity

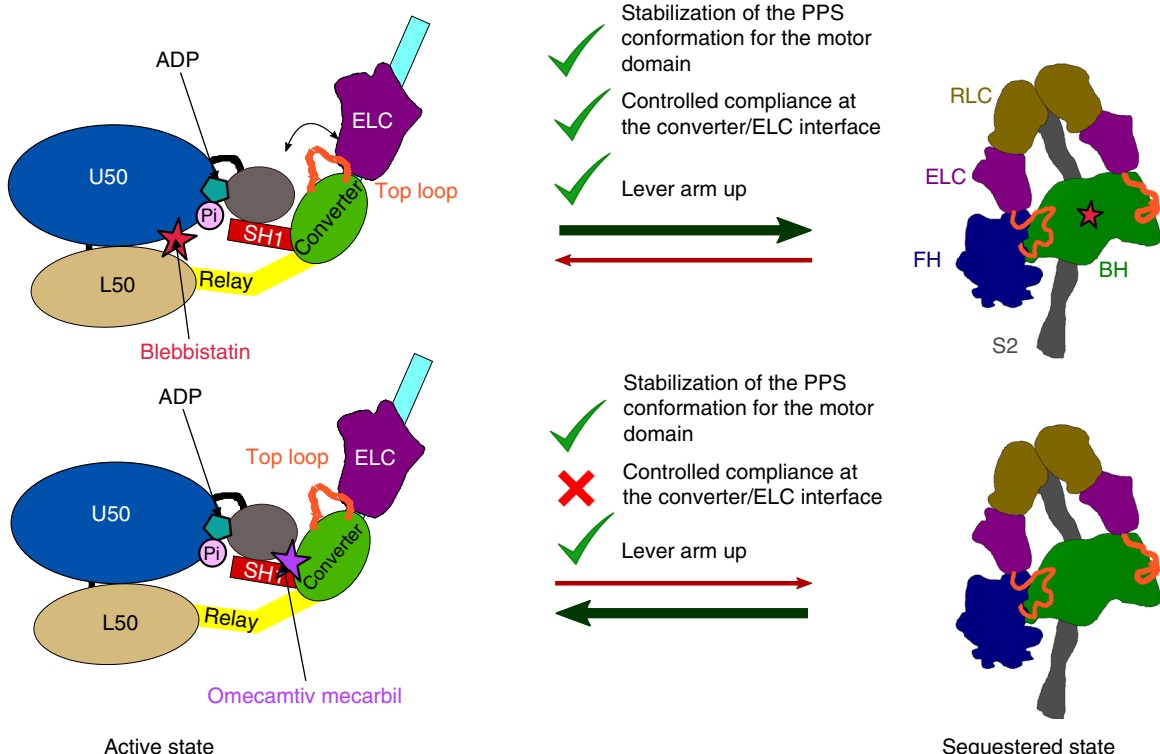

**Fig. 6** Antagonistic effects of omecamtiv mecarbil (OM) and blebbistatin (BS) on the sequestered state. Schematic representation of the effects of the inhibitor BS and the activator OM. On top: BS occupies a pocket within the motor domain core, close to the active site. Occupation of this pocket by BS allows to stabilize the PPS conformation without any effect on the local compliance of the lever arm required to form the sequestered state. Thus, BS binding is compatible with the IHM and favors the formation of the sequestered state. On the bottom: OM occupies a pocket at the interface between the converter and the N-ter subdomain[11]. This pocket is expected to decrease the compliance of the converter/ELC interface and constrain the lever arm in a particularly primed position. There is thus a loss in the compliance required to form the sequestered state if OM is bound to the heads of the Myo2 dimer. This explains why OM is incompatible with the formation of the sequestered state

for the heavy chain (MYH7), 95% for the ELC (MYL3), and 95% for the RLC (MYL2). We humanized the bovine model, performed minimization of the model and mapped the residues differing in sequence between the two species (Supplementary Fig. 7A). These polymorphisms are not located at the interfaces stabilizing the IHM and no difference is found between human and bovine IHM in which the interfaces between the BH and FH heads are conserved (Supplementary Fig. 7B).

Similarly, in order to be sure that the molecular dynamics results are transposable to the human β-cardiac myosin, we humanized the lever arm and performed molecular dynamics during 30 ns (Supplementary Fig. 7D). We observed similar dynamics on the human model in the top loop and with the musical chairs.

We then concluded that the human and the bovine β-cardiac myosins have the same behavior. What is observed from simulations of bovine β-cardiac myosin can be transposed with confidence to human β-cardiac myosin.

**Sequestered state model**. Two molecules of OM-PPS-S1 (PDB code: 56NA) and one molecule of cardiac S2 (based on a homology model with 2FXO, the human isoform, as a template) have been fitted in the 20 Å resolution electron density map obtained from Cryo-EM of thick filament of tarantula (Aphonopelma sp.; EMDB code EMD-1950) Flexor Metatarsus Longus striated muscle[52]. Fitting in electron density was performed with the function "fit in map" of USCF Chimera[53]. The RLC (bovine MYL2) and the second IQ motif were homology modeled based on squid isoforms (PDB code 3I5F) and fitted in the electron density. The model was made continuous and prepared with the CHARMM-GUI[54] with the Quick MD Simulator module and the CHARMM36[55] force field to describe the system in a box with explicit water (TIP3P model) and ions (KCl reaching 150 mM). The model was improved with several iterative steps of docking between the two heads performed with HADDOCK[37,56] and molecular dynamics without constraints (30 ns) on the complete model with Gromacs[57]. The only stage where interacting constraints were used was the HADDOCK docking and alternate possible assemblage modes were considered at this time. This docking stage thus allowed us to identify the best IHM model compatible with the low-resolution EM map.

The same procedure has been followed to fit our model in the 28 Å resolution electron density map from EM data of human cardiac thick filament[23] (EMDB code EMD-2240).

In previous studies, overall features of the sequestered state have been described from EM structures of tarantula leg striated muscle at a resolution of 20 Å, the best resolution to date[24,48] (PDB code 3JBH) and from those of human cardiac muscle at 28 Å[23] (EMDB code EMD-2240). Even if the thick filaments from these muscles have different helical geometries, the EM map and the structure of the two interacting heads are similar, although the resolution is not sufficient to discuss precisely the interfaces between the two heads. The sequestered model from tarantula striated muscle (PDB code 3JBH) has been obtained by a combination of homology modeling with the smooth muscle myosin sequestered state as a template (PDB code: 1I84), with flexible fitting and minimization in order to reduce steric clashes[24]. Recent models of the cardiac sequestered state have been obtained by homology modeling from the tarantula model, minimization and rigid body fit in tarantula and cardiac maps[32,35] (MS01 available on spudlab.stanford.edu; PDB code 5TBY). However, these models are not precise enough to discuss specifically the impact of a point mutation in an interface.

The molecular model we have built differs from those previously proposed MS01[35] and 5TBY[32]. The main difference is that our model has been computed from crystal structures describing the motor domain at atomic resolution (PPS bound to OM, PDB code: 5N69) and that molecular dynamics was used to refine the interfaces between the two heads. We are consequently confident that the modeled interactions between the FH and the BH correspond to a realistic and optimized interface for the IHM. This refined model is the only one to date to highlight the role of the top loop (in the FH head) for its interaction with the BH head. No bend in the lever arm is present for this refined model unlike what was previously proposed in other models (based on the previous use of the PPS atomic model of smooth muscle myosin (1BR1) in which a sharp bend occurs at the pliant region (Supplementary Fig. 6C) possibly due to a crystal packing artifact). To favor IHM intra-molecular interactions, a bend was introduced at the junction between the first and the second IQ (Supplementary Fig. 6A). The lack of atomic resolution structures for the cardiac RLC bound to the second IQ makes this region of the model less accurate. It has been modeled starting from a homology model of SqMyo2 (PDB code 3I5G). In the final steps of modeling, the S2 fragment was added. While the structure of this region is known, no precise information exists for its orientation: this is thus also a part of the model that would benefit in the future from higher-resolution data. In summary, the sequestered state model was built from atomic resolution crystal structures of the motor domain that allowed us to describe refined molecular contacts between the two myosin motor domains

after fitting of the model in the available EM maps and optimization of interactions (Supplementary Fig. 5A, B).

Note that two other models have been released by the Spudich lab: MS02 and MS03 (spudlab.stanford.edu). They both have been obtained with additional templates in order to correct some steric clashes and improve the model. MS01, MS02 and MS03 are very similar and share all the features discussed to be different from the model we have built in the study presented here, namely: the kink in the pliant region and the, non-canonical conformation of the IQs and of the RLC.

**Molecular dynamics for the lever arm simulations**. Molecular dynamics simulation inputs were prepared with the CHARMM-GUI[54,58] with the Quick MD Simulator module. The CHARMM36 force field[55] was used to describe the full systems in a box with explicit water (TIP3P model) and salt (KCl reaching 150 mM). The protocol provided in the Quick MD output was followed. Gromacs[57] (VERSION 5.0-rc1) was used to execute the simulations of 30 ns each. Each molecular dynamics simulation has been carried out multiple times (at least twice) in order to be sure that the results are reproducible and was further analyzed with the Gromacs tools.

**Mutations picking and prediction of the effects**. We analyzed a set of 178 HCM mutations (Supplementary Table 1, Supplementary Data 3) and 23 DCM mutations (Supplementary Data 4) occurring in the motor domain of β-cardiac myosin heavy chain (*MYH7*). Mutations were initially picked in expasy database (www.expasy.ch) and in recent publications[18,32,35]. For each mutation, we carefully checked the original publication that described the mutation in order to be sure that the mutation was strictly related to a HCM or a DCM diagnosis.

For each mutation, we made a prediction on the structural effect based on solid observations: (i) inspection of the IHM model to see if the mutation would alter the stability of the interfaces, (ii) molecular dynamics simulations performed for mutations from the converter in this study, (iii) previous knowledge on the mechanism controlling the recovery stroke transition and motor function in general. These observations allowed us to classify the mutations in six groups, depending on their effects. Structural/functional knowledge on myosin motor function was used to assess the ability of a particular mutation to affect the motor function based on previous experimental work on members of the myosin family. In the table, the expected function of the motor affected is mentioned. It is not possible from the structure to evaluate quantitatively how this mutation would modulate motor function—the table is, however, useful to distinguish mutations only affecting the stability of the off state with little effect on motor function (18%) from mutations in which both IHM stability and motor function is likely affected (16.5% + 31%). For the mutations we have used PPS stability affected when the prediction is very strongly thought to be correct; PPS stability likely affected for a good probability to have a strong effect—and PPS stability possibly affected for mutations for which the effect is likely but not certain. Some mutations for which an effect on the PPS stability and IHM formation is not predicted (mutations that may not be directly affecting the formation of the sequestered state) have been classified as protein stability and function altered when the current knowledge do predict a possible effect on motor function—and Motor function/stability mildly altered when the current knowledge of structural function is not strong enough to be sure whether a strong effect is to be expected: this mainly concerns the mutations in the N-terminal subdomain which has been recently proposed to play an important role for steps of the powerstroke (ADP release). The current data on these transitions is, however, too poor to make strong conclusions about the effect of mutations. To stay cautious on the interpretation of these mutations, a potential role on the stability of the myosin structure was also mentioned as it cannot be excluded. Percentages are calculated relative to the entire set of 178 HCM mutations that we analyzed.

## Data availability

The atomic coordinates and structure factors have been deposited in the Protein Data Bank, www.pdb.org, with accession number 6FSA (https://www.rcsb.org/structure/6FSA). The two models of the sequestered state of β-cardiac myosin are released in the Supplementary Information. The authors declare that all relevant data supporting the findings of this study are available on reasonable request.

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

## Acknowledgements

We thank James J. Hartman for providing the protein (Bovine cardiac fragment S1) and beamline scientists of ID23.2 (ESRF) for excellent support during data collection. We also thank Margaret A. Titus, Karl J. Petersen, Olena Pylypenko, and H. Lee Sweeney for critical reading of the manuscript. J.R.-P. was the recipient of an Association Française Contre les Myopathies (AFM) fellowship 18423. A.H. was supported by grants from CNRS, FRM DBI20141231319, AFM 17235, AFM 21805. The A.H. team is part of the Labex CelTisPhyBio:11-LBX-0038, which is part of the IDEX PSL (ANR-10-IDEX-0001-02 PSL).

## Author contributions

J.R.-P., D.A., and A.H. designed the research; J.R.-P. crystallized and solved the crystal structure; D.A. performed the modeling and molecular dynamics; A.H. classified the mutations and analyzed the consequences; all authors discussed and analyzed the data; J.R.-P., D.A., and A.H. wrote the manuscript.

## Additional information

**Competing interests:** The authors declare no competing interests.

