## [Peer Review File · Nature Communications]

Reviewers' Comments:

Reviewer #1:

Remarks to the Author:

This is a wonderful paper that describes a new structural model for the cardiac myosin head in the post rigor state. This leads to a much improved molecular model for the interacting heads motif that allows a reexamination of the causes of hypertrophic cardiomyopathies that fall into this region of myosin. Although there is perhaps too much speculation at times, the contents of this paper are timely and will be of great appeal to a wide audience. In particular it provides a more rigorous structural context for the ongoing discussion of the Spudich and Brenner models for HCM that is welcome at this time. Most importantly it will stimulate others to take a fresh look at this problem. A few minor comments are noted that will improve the presentation of this paper.

Line 102. The authors should check that abbreviations are defined as they arise ("PPS" is not defined on line 102). This will help the general reader. the authors should review the manuscript to ensure that abbreviations are defined or not used where possible.

Line 121 and later. The methods for the molecular dynamics are described to cursorily. More information is needed for the interested reader to assess the validity or scope of these calculations. This can be added to the supplementary material. In particular, how was the solvent handled? This has a profound effect on ionic interactions which are discussed extensively later in the manuscript and thus should be addressed directly. If the solvent was not considered then it will call into question how much weight to add to the existence or discussion of hydrogen bonds. Either way, this should be easy to address.

Figure 2D and following. The colors in this figure are very hard to follow. This will be particularly trying for those who suffer partial color blindness in particular deuteranomaly. This comment applies to other figures that follow. In any event increasing the color separation will aid the normal reader in following the contents of the figure.

The issue of colors also applies to the central column of Figure 3. Panel D is particularly hard to follow. In this figure it is very difficult to follow the hydrogen bonding pattern. The authors might consider putting stereo figures in supplementary material.

Figure 5 is a wonderful figure, but it is very difficult if not impossible to follow the colors in the current presentation. Even with a magnifying glass it is difficult. Given its importance to the overall outcome of this paper, the authors need to find a better color scheme or mode of presentation that allows the details and the big picture to come through.

Reviewer #2:

Remarks to the Author:

This is an ambitious paper from an experienced group. They have solved a new structure of bovine cardiac myosin motor domain in the post rigor conformation at 2.33 Å resolution and compared this with the previous pre-power stroke structure solved by the same group. This reveals an interesting pair of flexible loops in the converter domain that interacts with the ELC. Molecular dynamics shows these loops to be dynamic and they introduce the term – "musical chairs"- to describe the fluctuations between different binding partners in the two domains.

This work is solid and well described.

They go on to complete further molecular dynamics on mutations in the converter - ELC interface and mutations in the converter associated with HCM. They show that the mutations do cause alterations in the interaction between the flexible loops and the ELC and speculate on how these changes may alter the compliance of the motor domain and thus alter the mechanical behaviour of

the motor. I am not clear how the authors proceed from changes in flexibility to changes in muscle function – more details below.

The converter domain is also thought to be involved in the switched off, interacting heads structure seen in various thick filament structures. They use their crystal structure to fit into the low resolution EM images of thick filaments and produce a potential interface between the two interacting heads. This also favours a role for the converter flexible loops in making a contact between the two heads. This suggests a role for the myosin motor mutations in modulating the availability of myosin heads during contraction. These ideas are compatible with several recent studies suggesting a regulation of the number of available myosin heads in contraction, that this can be affected by a range of myosin mutations and potentially modulated by small molecule drugs. As such this observation is topical and could be important.

The authors then go a step further and survey all known HCM and DCM mutations and provide an explanation how each may affect motor function via effects on motor function, interacting heads, stability or a combination of effects. This is an long list and undoubtedly a lot of work to draw together and will be of great use to anyone working in the field. There is a huge amount of information in the paper and the experience eye of the lead author allows a certain trust in the inferences drawn. But it is not easy for the non-specialist to tell when solid information tips over into speculation. The paper feels like a combination of a strong structural paper and a review attempting to collate all the available data on cardiomyopathies by drawing together a vast array of literature. It is not clear to me how such definitive conclusions can be drawn from the data. There is no critical evaluation of the reliability of the data upon on which the cardiomyopathy data is evaluated.

Specific comments

Abstract. The abstract should make clear that the work is based on Bovine cardiac myosin.

Line 40/41

"To date the effect of HCM mutations on motor activity have not been precisely predicted." And quote ref 7 from 2014.

One of the reasons for this has been the non-availability of human cardiac myosin. Yet since 2014 there have been several studies of the mechanical properties of human cardiac motor domains primarily from the Spudich group carrying HCM & DCM mutations and several are quoted extensively (eg ref 31, 12, 37,38). The problem is not just knowing how the mutations alter the motor - but how this leads, over many years, to the disease.

Line 51 The authors start in the introduction by quoting the Brenner data in support of the idea that converter mutations could alter the stiffness of the converter and thus the myosin - and this idea is picked up at several places through the paper. Yet they quote of ref 31 at the end of the Introduction which sees little change in mechanical properties for mutations in the converter. The evidence for changes in stiffness by converter mutations is not clear cut. The authors need to clear on which side of the argument they sit. Both sides of the argument are repeated in the 2nd paragraph of the discussion Line 324-339.

L98 What is the homology between human and bovine cardiac motor domain and in the converter ELC specifically.

Line 103: Why was it necessary to build he ELC IQ ab initio

Line 140 through 199; Here the argument about the effect of converter mutations is developed. I found this section very confusing. What is the basis for the argument for which changes in structure cause a change in compliance of the motor and how this effects motor activity. This is not simple but the arguments are not clear and in some cases appear contradictory.

L146 "Large lever arm compliance resulting from such mutations would likely affect motor motor activity"

L77 " drastic loss of the controlled musical chairs dynamic at the converter/ELC interface leads to

more stiffness overall

L188 "The consequences on force production are likely moderate since the lever arm remains able to transmit and amplify the internal conformational changes of the motor.

Which mutation are predicted to alter compliance? Is there any experimental evidence that a significant effect on compliance is present. Or is the conclusion that the mutations are more likely to operate via the interacting heads conformer. The conclusion of this section is not clear to me. It is not clear to me how the predictive power of the ideas expressed have been or can be tested. At the simplest level there are many single site polymorphisms that are not associated with any disease. Can the approach used here test the lack of effect of these changes, ie it needs positive and negative controls?

P211. How reliable is a model built on a 20A Em map? This may be the best model to date but a lot of interpretation is built onto this detailed model.

P239 "A remarkable feature of the FH/BH model....." Is there any experimental evidence for the model? How can this be tested.

257 Implications for cardiac IHM model in disease.

The model can provide a powerful structural frame work to investigate the impact of HCM mutations but some evidence is required to know how reliable such a model is. A larger problem here is intrinsic to the whole field. We know from a whole host of experimental data, including from the author, that myosin is a very allosteric system. Perturbations in one part of the molecule can propagate to distinct or remote regions. It is often not possible to say that the effects of a mutation are local and confined.

The second problem for the field is that the mutations are present throughout adult life – if not throughout all life yet the disease is developed over 20-30 year in many cases. Thus the mutational trigger for disease development may be very mild. The dozen or so mutations that have been examined in detail for the human cardiac myosin have not shown obvious simple changes that translate into an explanation for the disease. The recent suggestion for HCM being linked to the interacting heads and alterations in availability of myosins is an exciting novel idea but remains largely untested. And as outline here mutations in the motor may have both affects on motor activity and availability of the myosin for interacting with actin. Mutation that affect myosin availability are unlikely to be all neutral for motor activity or stability.

Line 452/3 "all the crystal fragments to date are from bovine isoform..... Both isoforms" Both isoforms - are these not both the same isoform. And there are two human beta cardiac structures in the PDB data bank

Fig 2 B & C The numbering of residues in the Fig 2b do not seem to correspond to those in the MYH7 sequence in 2C. Is the numbering for human or bovine MyH7

Fig 5 How are mutations assigned to the 6 classes? What criteria are used?

Reviewer #3:

Remarks to the Author:

This paper involves a timely and exciting area of muscle and non-muscle biology dealing with what is currently being recognized as a global control mechanism for the myosin family of molecular motors. Here the focus is on beta-cardiac myosin and an Off-state known as the interacting heads motif (IHM). Existing models of the IHM for beta-cardiac myosin are homology models made from low-resolution EM images of tarantula skeletal muscle, and attempts to describe relevant side chain interactions to explain the effects of hypertrophic cardiomyopathy (HCM) mutations on the structure cannot be taken seriously. Numerous previous publications have, however, purported to be able to do so, potentially confusing the field. This paper emphasizes the need for a better IHM model and achieves that goal.

In this paper, the cardiac myosin converter/essential light chain interface is described for the first time from a 2.33 Å crystal structure of β-cardiac myosin, and an optimized quasi-atomic model of the sequestered state of cardiac myosin coupled to X-ray crystallography and in silico analysis of the mechanical compliance of the lever arm has allowed a much more realistic picture of the higher-resolution IHM structure. Their molecular model is likely superior to those previously proposed in that their model has been computed from crystal structures describing the motor domain head at atomic resolution and that molecular dynamics was used to refine the interfaces between the two heads.

Four of the well-known and severe HCM mutations located in the converter are analyzed in detail, and more than 100 HCM are characterized as to how they are likely to be affecting the biomechanical properties of the myosin and/or the IHM state. Importantly, the structures presented here provide complete atomic coordinates that are essential to evaluate the compliance of the lever arm and to study the consequences of specific missense mutations in this region. This work is well done, clearly written, and will make an important contribution to this important field. I recommend it for publication after the authors deal with the issues described below.

Issues that need to be addressed:

1. The authors need to provide their model as a pdb file (this was not provided to the reviewers) and they need in the manuscript to provide alignment data comparing their “better model” to the earlier models (2017 Alamo et al model published in eLife and earlier models MS01 and MS03 from the Spudich lab). In what regions are they very different and what regions are the models very similar. Are the major changes in the pliant region and light chain binding domains, as is true of MS03 and the Padron model? Are the light chains in a very different orientation from the earlier models? Is the proximal S2 tail in the same position relative to the myosin mesa or shifted significantly? The reader needs to know just how different these models really are in the various domains of the IHM structure. Furthermore, discussion is needed regarding differences and similarities between the detailed conclusions of side chain interactions proposed by Alamo et al in their 2017 eLife paper versus the conclusions presented here. Did Alamo et al get a lot right, in spite of having a model that according to the present authors should not have been good enough to make the predictions Alamo et al made?

2. The authors use their PPS crystal structure and molecular dynamics to model the IHM using existing low-resolution EM structures. The authors need to make clear that their model is just that, and that predictions about the interactions of specific amino acids are constrained by the low-resolution EM envelope into which they are docking their high-resolution crystal structure. The authors also claim that their approach results in a model that is significantly improved from previous attempts to model the IHM using other cardiac myosin crystal structures. They point out some key differences with respect to positioning of the lever arm and the interaction of the top loop of the FH converter domain with the BH, but for the latter it should again be made clear that this comparison results in testable hypotheses which, if proven correct, would then validate the claim that their modeling approach is more accurate than previous approaches.

In general, the authors need to be careful with statements such as:

Among these mutations, 18.5% of the set

272 represent mutations that destabilize the IHM with no significant effect on the motor
273 function (Fig. 5B, 5C). Some mutations that directly affect the IHM stability also have
274 effect on motor function (Fig. 5B, 5D, 16.5% of the set).

It needs to be clear throughout that these are predictions based on the authors' evaluation of the position of these residues within the structure, and not based on any biochemical/biomechanical data. This problem continues throughout the MS. It's great that the model is good enough to make

the predictions claimed, but they all need to be examined by appropriate biochemical/biophysical assays, and this needs to be made very clear early in the MS and in the discussion. This reviewer understands that such biochemical and biophysical experiments are beyond the scope of this manuscript.

Also:

"Interestingly, a
331 different point mutation at a particular position (R719W and R719G) can have
332 antagonistic effects on the lever arm pliancy. This illustrates how the sequence of this
333 region finely tunes the lever arm dynamics and validates the in silico approach to
334 investigate the effect of mutations on the dynamics of this region."

Again, the in silico approach can only be validated when the mutations have been studied in vitro and shown to support the predictions made based on the molecular dynamics studies. While R719W has been studied, there is no data on R719G that I am aware of (and none referenced in supp table 2). Furthermore, how do the authors explain that Kawana et al (in their reference list) found virtually no change in loaded motility between WT and R719W mutant myosin, which surely would have been altered by lever arm pliancy differences.

3. The authors map a large cohort of residues that have been associated with HCM onto their model and make predictions as to the effects of these mutations on the IHM, the stability of the PS, motor function, etc. While this is a very interesting exercise, it should be noted that not all of the listed mutations have been definitively shown to cause HCM and may be benign variants, thus complicating their findings. For example, the authors describe the putative effects of one mutation as an illustration:

"One of the mutations, A426T, occurs at a position that has been described in Myosin
292 VI (A422L) to slow the transition after Pi release. This mutation indeed introduces a
293 bulky residue in the 50 kDa cleft that must close during the powerstroke to bind actin
294 strongly. This indicates that some of these mutations can lead to HCM by altering
295 purely the states and transitions populated during the power stroke"

The potential problem is that A426T is not definitively known to be a disease causing variant. It is not in NCBI's ClinVar database, and indeed, the only reference given for it is a single publication that reports that a single patient with this variant AND a known pathogenic/likely pathogenic TNNI3 variant has HCM. So the authors are on shaky grounds to focus on this mutation. Similarly, all the mutations studied should be clearly pathogenic according to existing databases, and this should be looked at carefully throughout.

4. Species differences have been a problem in this area. Early work examined HCM mutations in the context of mouse alpha cardiac myosin, and the results suggested that ATPase, or velocity, or intrinsic force of the motor increased, supporting the view of hyper-contractility due to biomechanical changes. More recent work on human cardiac myosin gives different results (eg., Nag et al 2015 and Kawana et al 2017 references). The authors need to point this out and to make the point that the bovine cardiac myosin that they are studying is very close (how many residues different from human?) to the human and not expected to behave significantly different. They should emphasize that from 711-777 (in Fig 2 they could put both human and bovine sequences in and show that the numbering is identical and there is only one residue change S747G between human to bovine (note that in Fig 2C, 701 MYH7 should be 711)).

5. This manuscript reports on the post-rigor structure of bovine beta-cardiac myosin at 2.33A resolution and provides the first high-resolution look at the interface between the ELC and the converter domain.

Molecular dynamic simulations of this interface region for WT, as well as for 3 different mutations

within the converter domain, provide interesting insights into the potential role of this interface in normal WT function as well as in HCM-disease states. They make an interesting prediction as to the effect of the I736T mutation affecting the IHM state. For the other 3 converter domain mutations, the authors compare their predictions to the biochemical studies of these mutant proteins (Kawana et al in their reference list) but do not mention the fiber studies of these same mutations, which showed force increases of 1.3-1.8 fold rather than decreases in force as seen for the purified proteins. These fiber studies are already referenced in the paper but should be discussed in the context of the molecular dynamics simulations for these mutations.

Minor points:

1. Line 192: authors site the Kawana et al paper in reference to the statement "...mutations on fibers mechano-chemical parameters remain modest³¹". Did they mean to refer to the Kawana et al purified protein studies or rather to refer to fiber studies such as in references 13-15?

2. Line 310: "... (currently limited to a small set of mutations^{31,37,38}" omits references to Sommese et al (2013) and Bloemink et al (2014), references that are in sup table 2.

3. Fig 2 legend: Change G471R to G741R.

4. Fig 3: left panels, cannot see green sphere representing either R719W or R723G

5. Fig. 3: Authors point out that E732 and R783 increase the dynamics at the ELC/converter interface, but it is not clear from the putty diagram in Fig 3 that there are significant fluctuations at the R783 residue, nor do the authors explain what are the different conformations sampled by R783.

6. Fig 4: Some of the residue numbers in panel B are incorrect: L300 should be M300, M302 should be L302, D381 should be D382, L386 should be L387.

7. Supplementary tables 2 & 3 in their current form are extremely poorly formatted so as to make navigating them very frustrating, if not impossible - all 5 columns should be on the same page! As it is currently laid out, the reader must try to find the page with two columns (including refs) in the second half of the table that correspond to the 3 columns in the first half of the table. Also there should be a key explaining all terms (NPI, etc) as well as the color scheme for the text and the fill. The fill seems to correlate with the pie chart and I understand the coloring of the references based on the experimental system, but what do the colors of the text mean in the first 2 columns?

8. 269 "As previously proposed, SOME mutations are predicted to affect the IHM
270 stability since they impact residues found at the interfaces that stabilize the off
271 state^{33,30}". And

345 "Our results support the previous hypothesis that

346 SOME of the HCM mutations can disrupt the IHM and thus increase the number of
347 heads available to participate in force production¹⁷. This is the case for at least 65%
348 of the set we examined (Fig. 5B)."

References 17,30,33 proposed that MOST HCM mutations are likely to be destabilizing an off-state as a unifying hypothesis, in keeping with what these authors are concluding in this paper.

9. Transition between lines 55 and 56 – This seems the best place to bring reference 31 by Kawana et al to attention. Kawana et al used different methods from Brenner to assess the biomechanical properties of beta-cardiac myosin, and furthermore, for the first time, characterized the converter mutations in the context of purified human beta-cardiac myosin, where the HCM mutations occur. In the context of the human beta-cardiac myosin, Kawana et al found that a number of converter mutations resulted in only extremely small changes in ATPase activity,

velocity of contraction, and load-dependent force/velocity curves of WT vs mutant myosins. Significant changes in biomechanical properties of the lever arm should have been seen by the loaded in vitro motility assay in particular.

10. Line 84 – please insert here which mammalian cardiac myosin was used and how many residue differences there are between it and the human isoform.

11. 412 “Thus, stabilization of the pre-powerstroke state of the motor is not sufficient to
413 favor the myosin sequestered state and shutting down of motor activity during
414 relaxation. Discovery of drugs that directly affect or favor the formation of the IHM
415 motif would be of exquisite use to control more precisely the motors available.”

This relates to a point made by Anderson et al in a recent BioRxiv publication (Anderson, R.L., Trivedi, D.V., Sarkar, S.S., Henze, M., Ma, W., Gong, H., Rogers, C.S., Wong, F.L., Morck, M.M., Seidman, J.G., Ruppel, K.M., Irving, T.C., Cooke, R., Green, E.M and Spudich, J.A. (2018). Mavacamten stabilizes a folded-back sequestered super-relaxed state of β -cardiac myosin. bioRxiv <http://dx.doi.org/10.1101/266783>) where they propose that mavacamten, now in clinical trials by MyoKardia, causes the myosin S1 head to assume a conformation that favors the folded IHM state. Since Nature Communications allow references to bioRxiv papers, this paper could be referred to. This same paper can be referred to for the authors’ comments regarding the relationship between SRX and IHM on line 71. Anderson et al (bioRxiv <http://dx.doi.org/10.1101/266783>) have presented data with purified proteins confirming that the IHM state is in an SRX state.

Similarly, the Liu et al bioRxiv paper already referred to in this MS can be in the reference list as Liu, C., Kawana, M., Song, D., Ruppel, K.M. and Spudich, J.A. (2018) Controlling load-dependent contractility of the heart at the single molecule level. bioRxiv 258020; doi: <https://doi.org/10.1101/258020>.

Reviewers' comments:

Reviewer #1 (Remarks to the Author):

This is a wonderful paper that describes a new structural model for the cardiac myosin head in the post rigor state. This leads to a much improved molecular model for the interacting heads motif that allows a reexamination of the causes of hypertrophic cardiomyopathies that fall into this region of myosin. Although there is perhaps too much speculation at times, the contents of this paper are timely and will be of great appeal to a wide audience. In particular it provides a more rigorous structural context for the ongoing discussion of the Spudich and Brenner models for HCM that is welcome at this time. Most importantly it will stimulate others to take a fresh look at this problem. A few minor comments are noted that will improve the presentation of this paper.

Thank you for all these remarks, we worked on the figures in line with these advices in order to improve the presentation. We carefully have chosen colors to make them as easy to follow as possible and to be compatible with people suffering from color blindness. In these updated figures, we also changed the display of the top loop. In addition, major changes were introduced for Figure 5 to make it clearer and less redundant with table 1. We think that now the message provided by the figures is clearer and more visual.

Line 102. The authors should check that abbreviations are defined as they arise ("PPS" is not defined on line 102). This will help the general reader. the authors should review the manuscript to ensure that abbreviations are defined or not used where possible.

The abbreviations have been added. "Pre-powerstroke (PPS)"; "essential light chain (ELC)" (p.5); "root-mean-square deviation (RMSD)" (p.5).

Line 121 and later. The methods for the molecular dynamics are described to cursorily. More information is needed for the interested reader to assess the validity or scope of these calculations. This can be added to the supplementary material. In particular, how was the solvent handled? This has a profound effect on ionic interactions which are discussed extensively later in the manuscript and thus should be addressed directly. If the solvent was not considered then it will call into question how much weight to add to the existence or discussion of hydrogen bonds. Either way, this should be easy to address.

The details have been added to satisfy these requests in material and methods: "The CHARMM36 force field⁵⁵ was used to describe the full systems in a box with explicit water (TIP3P model) and salt (KCl reaching 150mM)." (p.23). We agree with the comments made about the solvent in simulations. In fact, we had used a model of solvent with explicit water molecules along with ions explicitly materialized (K⁺ and Cl⁻ present for both neutralizing the system net charge and reproduce the intracellular concentration). Note that periodic boundary conditions (PBC) were set to mimic an infinite system. In this context, the non-covalent bond exchanges observed during the simulations can be considered with confidence.

Figure 2D and following. The colors in this figure are very hard to follow. This will be particularly trying for those who suffer partial color blindness in particular deuteranomaly. This comment applies to other figures that follow. In any event increasing the color separation will aid the normal reader in following the contents of the figure.

The issue of colors also applies to the central column of Figure 3. Panel D is particularly hard to follow. In this figure it is very difficult to follow the hydrogen bonding pattern. The authors might consider putting stereo figures in supplementary material.

Figure 5 is a wonderful figure, but it is very difficult if not impossible to follow the colors in the current presentation. Even with a magnifying glass it is difficult. Given its importance to the overall outcome of this paper, the authors need to find a better color scheme or mode of presentation that allows the details and the big picture to come through.

All the figures were rethought and reconceptualized to gain in clarity, at least, so we hope. We have changed the colors so they could be more discernable and perceived by everyone according to a filter available in the software inkscape to simulate partial color blindness.

Reviewer #2 (Remarks to the Author):

This is an ambitious paper from an experienced group. They have solved a new structure of bovine cardiac myosin motor domain in the post rigor conformation at 2.33 Å resolution and compared this with the previous pre-power stroke structure solved by the same group. This reveals an interesting pair of flexible loops in the converter domain that interacts with the ELC. Molecular dynamics shows these loops to be dynamic and they introduce the term – “musical chairs”- to describe the fluctuations between different binding partners in the two domains. This work is solid and well described.

They go on to complete further molecular dynamics on mutations in the converter - ELC interface and mutations in the converter associated with HCM. They show that the mutations do cause alterations in the interaction between the flexible loops and the ELC and speculate on how these changes may alter the compliance of the motor domain and thus alter the mechanical behaviour of the motor. I am not clear how the authors proceed from changes in flexibility to changes in muscle function – more details below.

The converter domain is also thought to be involved in the switched off, interacting heads structure seen in various thick filament structures. They use their crystal structure to fit into the low resolution EM images of thick filaments and produce a potential interface between the two interacting heads. This also favours a role for the converter flexible loops in making a contact between the two heads. This suggests a role for the myosin motor mutations in modulating the availability of myosin heads during contraction. These ideas are compatible with several recent studies suggesting a regulation of the number of available myosin heads in contraction, that this can be affected by a range of myosin mutations and potentially modulated by small molecule drugs. As such this observation is topical and could be important.

The authors then go a step further and survey all known HCM and DCM mutations and provide an explanation how each may affect motor function via effects on motor function, interacting heads, stability or a combination of effects. This is an long list and undoubtedly a lot of work to draw together and will be of great use to anyone working in the field. There is a huge amount of information in the paper and the experience eye of the lead author allows a certain trust in the inferences drawn. But it is not easy for the non-specialist to tell when solid information tips over into speculation. The paper feels like a combination of a strong structural paper and a review attempting to collate all the available data on cardiomyopathies by drawing together a vast array of literature. It is not clear to me how such definitive conclusions can be drawn from the data. There is no critical evaluation of the reliability of the data upon on which the cardiomyopathy data is evaluated.

Thank you for all these useful and positive comments. We do agree that the main challenge with this study is to link the results from experimental coordinates and/or molecular dynamics to muscle function. Ideally, the structural information should be analyzed with functional data available for the mutations. The structures/models are indeed critical to interpret the results of functional studies. The fact that very little *in vitro* experimental data is available on mutations concerning HCM makes this study important since it can guide the set of mutations to be studied *in vitro* to gain insights about the disease.

Here we have chosen to study the converter mutations more particularly since a study recently published from the Spudich lab provided experimental data for these mutations and since our structures have provided coordinates for the converter/ELC interface.

Taking into account these comments, the description of the modeling has been modified to make it easier to follow for a non-specialist. It is very difficult to investigate the result of a mutation for motor compliance. We thus removed the hypothesis on how motor function would be affected by changes in the compliance, in the case of the test mutations R719G and R780E we had designed. These mutations have not been described as HCM mutations. The strength of the structural studies we have performed is to class the mutations in different types of possible impairments based on the structure and MD simulations. It is our hope that this study will be further tested by future functional investigations.

Regarding the humanized model of the IHM, we had in fact started the project but the results of the simulation have only recently been obtained. We have now decided to discuss in this paper the humanized model of the IHM in addition to the bovine model. Comparison of the two models shows that the interfaces of the IHM are not affected by the low polymorphism existing between the bovine and the human β -cardiac myosins. A 30 ns molecular dynamics of the human lever arm of β -cardiac myosin also indicates that the human lever arm displays similar dynamics to the bovine form (Supp. Fig. 6). This allowed us to conclude that the polymorphism present between the bovine and human β -cardiac myosin does not influence the dynamics of the lever arm or the IHM interfaces.

Specific comments

Abstract. The abstract should make clear that the work is based on Bovine cardiac myosin.

In fact, as mentioned above, the calculations for the human cardiac myosin have been completed and will be included. We now mention that the work is based on a structural study of Bovine cardiac myosin in the introduction: "Here, the cardiac myosin converter/essential light chain interface is described for the first time from a 2.33 Å crystal structure of bovine β -cardiac myosin (98% identity with the human form)." (p.4).

Line 40/41

"To date the effect of HCM mutations on motor activity have not been precisely predicted." And quote ref 7 from 2014.

One of the reasons for this has been the non-availability of human cardiac myosin. Yet since 2014 there have been several studies of the mechanical properties of human cardiac motor domains primarily from the Spudich group carrying HCM & DCM mutations and several are quoted extensively (eg ref 31, 12, 37,38). The problem is not just knowing how the mutations alter the motor - but how this leads, over many years, to the disease.

The idea of development over time of the disease has been added, we modified the sentence to "To date, the effects of HCM mutations on motor activity and how they can lead to such a disease over the years have not been precisely deciphered" (p.2).

Line 51 The authors start in the introduction by quoting the Brenner data in support of the idea that converter mutations could alter the stiffness of the converter and thus the myosin - and this idea is picked up at several places through the paper. Yet they quote of ref 31 at the end of the Introduction which sees little change in mechanical properties for mutations in the converter. The evidence for changes in stiffness by converter mutations is not clear cut. The authors need to clear on which side of the argument they sit. Both sides of the argument are repeated in the 2nd paragraph of the discussion Line 324-339.

In order to clarify this point, we added a sentence in discussion line 368: "Then, according to our results, even if the R719W and R723G mutations induce a small decrease in force production of individual heads³¹, they also destabilize the sequestered state and increase the number of heads available to produce force, bringing a possible explanation for why fibers from patients carrying these mutations produce more force^{13,14,15}." (p.16).

It is difficult to make a link between force production of a fiber and the motor function impairment of individual heads, because the fiber is a complex system. In his studies, Brenner concludes that (i) since the converter swings during the motor cycle, it is likely to be essential in regulating stiffness; (ii) HCM mutations on the converter increase the stiffness and force produced by the fibers. In fact, we know from recent studies that R723G and R719W slightly reduce the force produced by the recombinant cardiac myosin S1 fragment (Kawana *et al.*, 2017). So the increase of force production seen in fibers carrying these mutations is likely due to a destabilization of the sequestered state that increases the number of heads available to produce force. This point has also been raised in the specific comments of reviewer #3 (comment 5).

L98 What is the homology between human and bovine cardiac motor domain and in the converter ELC specifically.

The identity between human and bovine cardiac myosins is 98% for the heavy chain and 95% for the ELC. We have added to this study the human model of the IHM that has been minimized (see Material and Methods p. 19-20 in the section "Human β -cardiac myosin"; Supp. Fig. 6A and B). This model shows that

the few polymorphic residues between bovine and human cardiac myosins are far from the converter/ELC interface and from the interfaces between heads that stabilize the sequestered state. Thus the two models of the IHM structure are very similar after minimization. As expected from the EM data available from different organisms and different myosins, the IHM structure is quite conserved. We therefore chose to work on bovine cardiac myosin since we had the coordinates for this motor. The main strength of this work is to provide a quasi-atomic model obtained from a crystal structure of the motor domain that is an orthologue with "98% identity".

Line 103: Why was it necessary to build the ELC IQ *ab initio*

We built this part of the model *ab initio* since there was no structure coordinates available. Eventually, independent modelling/building of this region in (PPS-omecantiv and PR cardiac S1 structures) ended up building very similar models for this essential part of the motor.

Line 140 through 199; Here the argument about the effect of converter mutations is developed. I found this section very confusing. What is the basis for the argument for which changes in structure cause a change in compliance of the motor and how this effects motor activity. This is not simple but the arguments are not clear and in some cases appear contradictory.

We took into account these comments and tried to make the argument clearer and more organized. We removed the arguments that are too speculative, especially those linking compliance to force production for test mutations. We now discuss changes observed by introducing mutations as modulation of the structural plasticity that would be required for the formation of the IHM. All the changes appear in the paragraph "**The cardiac lever arm internal dynamics**" (p. 6-7).

L146 "Large lever arm compliance resulting from such mutations would likely affect motor motor activity"

We agree, this sentence is speculative and could be misleading. We thus removed this sentence from the text.

L77 "drastic loss of the controlled musical chairs dynamic at the converter/ELC interface leads to more stiffness overall

L188 "The consequences on force production are likely moderate since the lever arm remains able to transmit and amplify the internal conformational changes of the motor.

Which mutation are predicted to alter compliance? Is there any experimental evidence that a significant effect on compliance is present. Or is the conclusion that the mutations are more likely to operate via the interacting heads conformer. The conclusion of this section is not clear to me. It is not clear to me how the predictive power of the ideas expressed have been or can be tested. At the simplest level there are many single site polymorphisms that are not associated with any disease. Can the approach used here test the lack of effect of these changes, ie it needs positive and negative controls?

We edited the text to clarify this point (p. 8-9, section "**Converter HCM mutations alter the lever arm internal dynamics via long-range effects on a global charge network at the converter/ELC interface**"). While the simulations provide information about the rigidity/structural plasticity of the lever arm, this cannot be fully translated to predict the effect on force production. This approach provides predictions for how different mutations can restrict or increase structural dynamics overall in the lever arm. In addition, an important result of our study is that the dynamics at the converter/ELC interface is required for the formation of the sequestered state. This can be tested by investigating whether these mutations would indeed increase the number of active heads. We have now removed the speculative statements about compliance and effect on motor function. We have kept however the discussion of the effect of these mutations on the sequestering of the heads.

The text was also clarified to show that we investigated two types of mutations: **(i)** two test mutations introduced in the anchoring positions (R719G and R780E). These mutations induce a large increase in the compliance of the lever arm and **(ii)** four HCM mutations, three of which (R719W, R723G, G741R) are slightly stiffening the lever arm.

According to the results from Kawana *et al.*, 2017, these three mutations have no significant effect on

mechano-chemical properties, so we can conclude that the slight decrease in structural plasticity has no major effect on motor function.

The conclusions of our experiments is that (i) we understood how this charge network at the ELC/converter interface can control the compliance of the lever arm and (ii) the main effects of HCM mutations is a modification in the top loop conformation. We describe in addition that this top loop is a major part of the IHM interface and that the changes in conformation and dynamics of this region can have major effect in the sequestered state stability, which is in favor of the hypothesis previously proposed by Jim Spudich.

The approach used here has allowed us to get both positive and negative controls:

- (i) We can clearly see it with the mutation I736T which has no effect on the structural dynamics of the lever arm, this could be considered as a negative control.
- (ii) We added to the manuscript a 30 ns molecular dynamics carried on the humanized lever arm of β -cardiac myosin. We observe similar dynamics of the “musical chairs” and the top loop (Supp. Fig. 6D). This experiment was meant to check that the humanization of the structure does not induce any major change in structural plasticity, dynamics, and finally this validates our approach.

P211. How reliable is a model built on a 20Å Em map? This may be the best model to date but a lot of interpretation is built onto this detailed model.

P239 “A remarkable feature of the FH/BH model.....” Is there any experimental evidence for the model? How can this be tested.

The structures and models described here allows us to gain significant improvement in the description of the motor domain and the way they interact with one another in the sequestered state. Our model is improved because it is based on the structure of cardiac myosin in the pre-powerstroke state (PPS). It was indeed proposed from previous EM data that the sequestered state includes two heads in this PPS. Both heads of the IHM are in a state with the lever arm up. The fact that Blebbistatin (that favors PPS) stabilizes the sequestered state is reinforcing this proposition. The major strength of our model compared to those proposed previously is that it is based on crystal structures that provide very precise definition of the structure of the motor domain and the converter/ELC interface. Based on this, the molecular dynamics allowed us to model molecular contacts for the sequestered state and to probe the dynamics of the converter/ELC interface, we thus are able to discuss realistic interfaces between heads. This was not which were not possible from previous models based on non-cardiac myosin coordinates docked as flexible body into a 20 Ångströms resolution map. While we agree that a higher resolution model is required for accurate coordinates of the sequestered state, our model is superior and sufficient to describe precisely interactions between the heads and to analyze precisely the effect of mutations within the motor domain and in particular the converter.

It is clear that the IHM structure proposed in this work remains a model and that future investigations and higher resolution structures will allow to go further in the discussion and to confirm/edit the model. Recently, Jim Spudich's group has been able to produce purified recombinant HMM and to evaluate the impact of mutations and drugs on the proportion of heads folded in the sequestered state (Anderson *et al.*, 2018). A good method to test our model would be to measure the direct impact of the I736T mutation on myosin motor function and on the proportion of molecules able to adopt the sequestered state. Our analysis predicts that the motor function should not be significantly altered for this I736T mutant but that the mutation destabilizes drastically the sequestered state. Similar studies with the R719W mutant should also confirm the importance of the plasticity in converter/ELC interface for the formation of the sequestered state. We hope that this work will provide a structural framework and give guidelines for scientists in the field for future investigations that will allow to progress on the study of HCM and the auto-inhibited state.

257 Implications for cardiac IHM model in disease.

The model can provide a powerful structural framework to investigate the impact of HCM mutations but some evidence is required to know how reliable such a model is. A larger problem here is intrinsic to the whole field. We know from a whole host of experimental data, including from the author, that myosin is a very allosteric system. Perturbations in one part of the molecule can propagate to distinct or remote regions. It is often not possible to say that the effects of a mutation are local and confined. The second problem for the field is that the mutations are present throughout adult life – if not

throughout all life yet the disease is developed over 20-30 year in many cases. Thus the mutational trigger for disease development may be very mild. The dozen or so mutations that have been examined in detail for the human cardiac myosin have not shown obvious simple changes that translate into an explanation for the disease. The recent suggestion for HCM being linked to the interacting heads and alterations in availability of myosins is an exciting novel idea but remains largely untested. And as outline here mutations in the motor may have both affects on motor activity and availability of the myosin for interacting with actin. Mutation that affect myosin availability are unlikely to be all neutral for motor activity or stability.

We agree with the reviewer that the structural insights gained from our study must be completed with functional studies. While it is not possible to predict in detail how HCM mutations can impact the motor function, we were able to define classes that will guide further choices for the study of the mutations. The structural model and our predictions will thus be tested precisely. Once these validations obtained, we hope that such classes of mutations could also guide understanding of the disease progression and hopefully also guide decisions towards treatment. Previous investigations on the recombinant S1 fragment have already demonstrated that three mutations of the converter (R719W, R723G and G741R) induce no major change in motor function (Kawana *et al.*, 2017) but pediatric mutations such as D239N and H251N have more drastic effects on force and ATPase (Adhikari *et al.*, 2016). We predict that D239N, which is located in switch I and plays a role in nucleotide binding, would alter both the stability of the IHM and motor function since it would alter the stability of the PPS, thus modifying the stability of the IHM in addition to alter motor function. These examples are a proof-of-concept that the model presented here is consistent with *in vitro* studies and helps the interpretation of these results. Our analysis of the sequestered state structure allows us to predict whether destabilization of the IHM is likely for the diverse HCM mutations identified to date. We also provide the nature of the destabilization: whether it occurs via destabilization of the interactions between the myosin heads, or whether it occurs due to destabilization of the PPS state. Here we indicate clearly which mutations might have both an effect on motor function and IHM stability. These predictions will guide the functional assays to be performed and the structural insights gained by our studies will thus be easily tested and challenged by the availability of increasing amount of data from *in vitro* studies. Our analysis clearly indicate that the disease is complex and variable depending on the mutation but also starts to provide ways to organize explain the extreme complexity and variability of this pathology. After validation with functional assays, regrouping mutations into classes will help decision of the assays to be performed for therapeutic treatment with drugs for example.

Line 452/3 “all the crystal fragments to date are from bovine isoform..... Both isoforms” Both isoforms - are these not both the same isoform. And there are two human beta cardiac structures in the PDB data bank

Thank you for this comment. The statement was wrong and has been corrected: it is the same isoform but from different organisms, now the text is: “In this work, we chose to use the bovine β -cardiac myosin.” (p.19). There are indeed human beta-cardiac structures in the PDB (4P7H, 4DB1 and 4PA0). Unfortunately, these structures include only the motor domain without the ELC and they are in a post-rigor state that is different from the pre-powerstroke state used to model the IHM in which both heads have the lever arm up. These PDB structures were not proper for the study we performed here: they cannot allow the investigation of the converter/ELC interface and they cannot be used to model the IHM.

Fig 2 B & C The numbering of residues in the Fig 2b do not seem to correspond to those in the MYH7 sequence in 2C. Is the numbering for human or bovine MyH7

The sequence is from bovine MyH7 but there was a mistake in numbering, this has been corrected.

Fig 5 How are mutations assigned to the 6 classes? What criteria are used?

The evaluation of the effect of mutations was carefully assessed from a combination of structural and functional knowledge on the myosin motor. While no prediction can replace fully a quantitative measure, these predictions are based on solid observations:

- inspection of the ‘sequestered-state’ model built within this study for accounting on the interactions between heads
- Molecular dynamics simulations performed in this study for mutations in the converter and their impact on the dynamics of the lever arm

- Knowledge of the mechanism controlling the recovery stroke transition and the factors that are important for the stability of the pre-powerstroke state based on previous structural/functional data acquired for myosin motors. A description of why a mutation would lead to the destabilization of the pre-powerstroke state is given for each mutation in the **Supp. Table 2**. While the effect of the mutation is likely, it cannot always be predicted whether this effect would be a major one (i.e. whether it would prevent the formation of the sequestered state because the population of heads in the primed lever arm position adequate for forming the sequestered state would become a minority) – or if this effect would be a minor one (leading to only a little decrease of heads in the sequestered state). Anyway, one of the messages of the paper is that we predict that a number of HCM mutations affect the number of heads able to adopt the sequestered state mainly due to an effect on destabilization of the primed PPS conformation that is a pre-requisite for the heads to start to interact and form the sequestered state. For the mutations we have used ‘PPS stability affected’ when the prediction is very strongly thought to be correct; ‘PPS stability likely affected’ for a good probability to have a strong effect – and ‘PPS stability possibly affected’ for mutations for which the effect is likely but not certain.
- Structural/functional knowledge on myosin motor function was used to assess the ability of a particular mutation to affect the motor function based on previous experimental work on members of the myosin family. In the table, the expected function of the motor affected is mentioned. It is not possible from the structure to evaluate quantitatively how this mutation would modulate motor function – the table is however useful to distinguish mutations only affecting the stability of the off state with little effect on motor function (18.5%) from mutations in which both IHM stability and motor function is likely affected (16.5% + 30.5 %).
- Some mutations for which an effect on the PPS stability and IHM formation is not predicted (mutations that may not be directly affecting the formation of the sequestered state) have been classified as ‘protein stability and function altered’ when the current knowledge do predict a possible effect on motor function – and ‘Motor function/stability mildly altered’ when the current knowledge of structural function is not strong enough to be sure whether a strong effect is to be expected : this mainly concerns the mutations in the N-terminal subdomain which has been recently proposed to play an important role for steps of the powerstroke (ADP release). The current data on these transitions is however too poor to make strong conclusions about the effect of mutations. To stay cautious on the interpretation of these mutations, a potential role on the stability of the myosin structure was also mentioned as it cannot be excluded.

We have now added this information in material and methods pages 23-24-25: “For each mutation, we made a prediction on the structural effect based on solid observations: (i) inspection of the IHM model to see if the mutation would alter the stability of the interfaces (ii) molecular dynamics simulations performed for mutations from the converter in this study, (iii) previous knowledge on the mechanism controlling the recovery stroke transition and motor function in general. These observations allowed us to classify the mutations in six groups, depending on their effects. Structural/functional knowledge on myosin motor function was used to assess the ability of a particular mutation to affect the motor function based on previous experimental work on members of the myosin family. In the table, the expected function of the motor affected is mentioned. It is not possible from the structure to evaluate quantitatively how this mutation would modulate motor function – the table is however useful to distinguish mutations only affecting the stability of the off state with little effect on motor function (18.5%) from mutations in which both IHM stability and motor function is likely affected (16.5% + 30.5 %). For the mutations we have used ‘PPS stability affected’ when the prediction is very strongly thought to be correct; ‘PPS stability likely affected’ for a good probability to have a strong effect – and ‘PPS stability possibly affected’ for mutations for which the effect is likely but not certain. Some mutations for which an effect on the PPS stability and IHM formation is not predicted (mutations that may not be directly affecting the formation of the sequestered state) have been classified as ‘protein stability and function altered’ when the current knowledge do predict a possible effect on motor function – and ‘Motor function/stability mildly altered’ when the current knowledge of structural function is not strong enough to be sure whether a strong effect is to be expected : this mainly concerns the mutations in the N-terminal subdomain which has been recently proposed to play an important role for steps of the powerstroke (ADP release). The current data on these transitions is however too poor to make strong conclusions about the effect of mutations. To stay cautious on the interpretation of these mutations, a potential role on the stability of the myosin structure was also mentioned as it cannot be excluded. Percentages are calculated relative to the entire set of 179 HCM mutations that we analyzed”.

Reviewer #3 (Remarks to the Author):

This paper involves a timely and exciting area of muscle and non-muscle biology dealing with what is currently being recognized as a global control mechanism for the myosin family of molecular motors. Here the focus is on beta-cardiac myosin and an Off-state known as the interacting heads motif (IHM). Existing models of the IHM for beta-cardiac myosin are homology models made from low-resolution EM images of tarantula skeletal muscle, and attempts to describe relevant side chain interactions to explain the effects of hypertrophic cardiomyopathy (HCM) mutations on the structure cannot be taken seriously. Numerous previous publications have, however, purported to be able to do so, potentially confusing the field. This paper emphasizes the need for a better IHM model and achieves that goal.

In this paper, the cardiac myosin converter/essential light chain interface is described for the first time from a 2.33 Å crystal structure of β -cardiac myosin, and an optimized quasi-atomic model of the sequestered state of cardiac myosin coupled to X-ray crystallography and in silico analysis of the mechanical compliance of the lever arm has allowed a much more realistic picture of the higher-resolution IHM structure. Their molecular model is likely superior to those previously proposed in that their model has been computed from crystal structures describing the motor domain head at atomic resolution and that molecular dynamics was used to refine the interfaces between the two heads.

Four of the well-known and severe HCM mutations located in the converter are analyzed in detail, and more than 100 HCM are characterized as to how they are likely to be affecting the biomechanical properties of the myosin and/or the IHM state. Importantly, the structures presented here provide complete atomic coordinates that are essential to evaluate the compliance of the lever arm and to study the consequences of specific missense mutations in this region. This work is well done, clearly written, and will make an important contribution to this important field. I recommend it for publication after the authors deal with the issues described below.

We thank the reviewer for these positive comments and for the minor points he mentioned. Most of these minor points were mistakes and they have now been corrected.

Issues that need to be addressed:

1. The authors need to provide their model as a pdb file (this was not provided to the reviewers) and they need in the manuscript to provide alignment data comparing their “better model” to the earlier models (2017 Alamo et al model published in eLife and earlier models MS01 and MS03 from the Spudich lab). In what regions are they very different and what regions are the models very similar. Are the major changes in the pliant region and light chain binding domains, as is true of MS03 and the Padron model? Are the light chains in a very different orientation from the earlier models? Is the proximal S2 tail in the same position relative to the myosin mesa or shifted significantly? The reader needs to know just how different these models really are in the various domains of the IHM structure. Furthermore, discussion is needed regarding differences and similarities between the detailed conclusions of side chain interactions proposed by Alamo et al in their 2017 eLife paper versus the conclusions presented here. Did Alamo et al get a lot right, in spite of having a model that according to the present authors should not have been good enough to make the predictions Alamo et al made?

We will release the coordinates of both the bovine and humanized IHM models that were built for this study (Mat&Met).

In Supp. Fig. 5, we compared in detail the IHM model presented here and earlier models discussed in publications (MS01 from Spudich’s lab and 5TBY from Padron’s lab). As shown in Supp. Fig. 5A, a major difference between models is located in **the lever arm** where there is a kink in the pliant region in previous models, probably because they were built from the crystal structure of the smooth muscle myosin (Wendt *et al.*, 2001). This kink in fact is likely an artefact of the crystal packing of this 1BR1 structure. The kink has not been observed for other myosin structures. A major consequence of this difference is a modification of **the ELC/converter interface** since the converter is rotated 25° compared to the ELC position because of the kink, even if the conformation of the ELC is similar in all models (Supp. Fig. 5C). In addition, the **RLC** is very different in our model (Supp. Fig. 5D), especially in the N-lobe position which is not bound to a helical part of

the heavy chain in previous models. To improve this part of the model, we have built a homology model with the RLC structure as found in the squid myosin 2 S1 structure as a template (PDB code: 3I5G). The interface between heads is also different in the model presented here. We illustrate it with the difference in the contribution of the top loop from the FH which is in different conformations in all the models (Supp. Fig. 5E). The S2 conformation is also different in all models (Supp. Fig. 5B). This part of the molecule is the one in which we have less confidence and the molecular contacts of the S2 and the BH-mesa while stable in our simulations are hard to model in detail without further structural information. More generally, the interfaces from the IHM are also very different in our model compared to previous models. This is linked to the method used in previous models which were obtained via homology modeling and flexible fitting in the electron density map. This approach causes major deformations of the myosin structure in the motor domain as well as in the LCs and the lever arm.

All these discussions about the differences between the model have been developed in Mat&Met, section "Sequestered state model" (p.20-21-22) and a reference of this section was introduced in the main text (lines 212-216). We changed the text which is now: "Since the human and the bovine β -cardiac myosins are very close in sequence identity, we also humanized the bovine model (see Methods). The polymorphism between these two models does not lead to any difference for the interfaces that stabilize the IHM (see Sup Fig 6A, 6B). The model perfectly fits in current low-resolution available maps (Supp. Fig. 4). This model is the best to date for two reasons (see Methods for a detailed comparison with previous models)" (p.9-10). This will guide the reader to the Methods section for a detailed comparison of the models (section "Sequestered state model", p.20-21-22). For reasons of presentation and clarity we only compare our model with MS01 in the figure, but even if MS03 slightly differs in the orientation of the lever arm and in the conformation of the LCs, it has also a kink in the pliant, an inappropriate conformation of the RLC and a deformation of the motor domain. So the conclusions regarding the comparison with our model are the same.

We added a paragraph at the end of the section "sequestered state model" (p.23) in order to make a precision on this point: 'Note that two other models have been released by the Spudich lab: MS02 and MS03 (spudlab.stanford.edu). They both have been obtained with additional templates in order to correct some steric clashes and improve the model. MS01, MS02 and MS03 are very similar and share all the features discussed to be different from the model we have built in the study presented here, namely: the kink in the pliant region and the, non-canonical conformation of the IQs and of the RLC.'

The model from Alamo *et al.*, 2017 has two main differences compared to the model proposed here (i) it has been obtained by homology with a model (Tarantula myosin II) that has been fitted in the density with flexible fit, so the protein fold is deformed and (ii) the molecular contact have not been refined. A difference of accuracy thus occurs between the two models. Alamo *et al.* only discussed the positions of residues without much accuracy on whether they are exposed or buried in the domains they belong. They mentioned whether the mutations would be close enough to the region of interaction (as defined in the paper on tarantula IHM) or in a subdomain that can be important for motor function (for example the relay helix). In the end, their description is not wrong but they could not distinguish whether the residues would have an impact on structure stability, function or stabilization of direct interactions between heads. The inherent characteristics of their model do not allow them to discuss with the same accuracy as we do: this was sufficient to highlight that some residues could have an impact on the IHM stability but not to analyze and propose that some mutations would also have an impact on motor function or that mutations destabilizing the pre-powerstroke state could impact the IHM stability.

2. The authors use their PPS crystal structure and molecular dynamics to model the IHM using existing low-resolution EM structures. The authors need to make clear that their model is just that, and that predictions about the interactions of specific amino acids are constrained by the low-resolution EM envelope into which they are docking their high-resolution crystal structure. The authors also claim that their approach results in a model that is significantly improved from previous attempts to model the IHM using other cardiac myosin crystal structures. They point out some key differences with respect to positioning of the lever arm and the interaction of the top loop of the FH converter domain with the BH, but for the latter it should again be made clear that this comparison results in testable hypotheses which, if proven correct, would then validate the claim that their modeling approach is more accurate than previous approaches.

In general, the authors need to be careful with statements such as:

Among these mutations, 18.5% of the set
272 represent mutations that destabilize the IHM with no significant effect on the motor
273 function (Fig. 5B, 5C). Some mutations that directly affect the IHM stability also have
274 effect on motor function (Fig. 5B, 5D, 16.5% of the set).

It needs to be clear throughout that these are predictions based on the authors' evaluation of the position of these residues within the structure, and not based on any biochemical/biomechanical data. This problem continues throughout the MS. It's great that the model is good enough to make the predictions claimed, but they all need to be examined by appropriate biochemical/biophysical assays, and this needs to be made very clear early in the MS and in the discussion. This reviewer understands that such biochemical and biophysical experiments are beyond the scope of this manuscript.

Also:

"Interestingly, a
331 different point mutation at a particular position (R719W and R719G) can have
332 antagonistic effects on the lever arm pliancy. This illustrates how the sequence of this
333 region finely tunes the lever arm dynamics and validates the *in silico* approach to
334 investigate the effect of mutations on the dynamics of this region."

Again, the *in silico* approach can only be validated when the mutations have been studied *in vitro* and shown to support the predictions made based on the molecular dynamics studies. While R719W has been studied, there is no data on R719G that I am aware of (and none referenced in supp table 2). Furthermore, how do the authors explain that Kawana et al (in their reference list) found virtually no change in loaded motility between WT and R719W mutant myosin, which surely would have been altered by lever arm pliancy differences.

Based on these comments and comments from reviewer #2, we edited the section "The cardiac lever arm internal dynamics" (p.5-6-7) where the results of the molecular dynamics are analyzed in order to clarify it. We now clearly state that we have investigated two kinds of mutations, some are HCM mutations but others are "test mutations" (R719G and R780E), which have not been associated with the disease. We have chosen these two converter mutations to test how these anchoring residues would affect structural plasticity/compliance of the lever arm. Our molecular dynamics experiments indicate that these mutations increase the structural plasticity of the lever arm. To describe this result, the text has been edited to avoid speculation and mainly describe the impact of these mutations in structural plasticity of the converter/ELC interface.

We also carried out the dynamics on three HCM mutations from the converter (R719W, R723G and G741R) that in fact increase slightly the stiffness of the lever arm (according to the dynamics) and modify the dynamics of the top loop. Previous results indicate that these mutations have no significant effect on motor function (Kawana *et al.*, 2017). The results from Kawana *et al.*, 2017 demonstrate that R719W and R723G only resulted in a small decrease in force production, in line with our conclusions. After inspection of the IHM model, we conclude that the major effect of these small changes in compliance and dynamics are impacting the formation of the sequestered state since they impact the Converter/ELC interface and the top loop conformation which are critical for the formation of the sequestered state.

3. The authors map a large cohort of residues that have been associated with HCM onto their model and make predictions as to the effects of these mutations on the IHM, the stability of the PS, motor function, etc. While this is a very interesting exercise, it should be noted that not all of the listed mutations have been definitively shown to cause HCM and may be benign variants, thus complicating their findings. For example, the authors describe the putative effects of one mutation as an illustration:

"One of the mutations, A426T, occurs at a position that has been described in Myosin
292 VI (A422L) to slow the transition after Pi release. This mutation indeed introduces a
293 bulky residue in the 50 kDa cleft that must close during the powerstroke to bind actin
294 strongly. This indicates that some of these mutations can lead to HCM by altering
295 purely the states and transitions populated during the power stroke"

The potential problem is that A426T is not definitively known to be a disease causing variant. It is not in NCBI's ClinVar database, and indeed, the only reference given for it is a single publication that reports that a single patient with this variant AND a known pathogenic/likely pathogenic TNNI3 variant has HCM. So the authors are on shaky grounds to focus on this mutation. Similarly, all the mutations studied should be clearly pathogenic according to existing databases, and this should be looked at carefully throughout.

It is right that the mutant A426T is present in a patient quite young (around 30) that also carries a mutation in TNNI3. On the other hand, this patient has been clearly diagnosed as carrying HCM. It is possible that the two mutations participate in setting up HCM. The presence of the mutation in TNNI3 does not exclude the fact that the mutation in the myosin head has no impact. In fact, earlier data on myosin function strongly suggest that this mutation will have an effect on the powerstroke. Introduction of a bulky residue has been shown to impact the powerstroke but not Pi release *in vitro* for myosin VI, myosin V and Myosin II (Llinas *et al.*, 2015). These previous data support the fact that this residue would be part of the small class of mutations (7.5% of mutations) that are predicted to alter only the myosin motor function). The classification table allows to highlight such mutations which are interesting to investigate to show that HCM can result from increase in force production without any effect on the number of motors activated (no destabilization of the sequestered state). In the case of A426T, functional data but also evaluation of the evolution of the HCM disease in animal models are needed. However, it is an interesting one to study since there is a clear prediction that this mutation would not affect the IHM stability and thus it would be a way to test whether an impact on motor function (possible increase in force production) could by itself cause HCM.

4. Species differences have been a problem in this area. Early work examined HCM mutations in the context of mouse alpha cardiac myosin, and the results suggested that ATPase, or velocity, or intrinsic force of the motor increased, supporting the view of hyper-contractility due to biomechanical changes. More recent work on human cardiac myosin gives different results (eg., Nag et al 2015 and Kawana et al 2017 references). The authors need to point this out and to make the point that the bovine cardiac myosin that they are studying is very close (how many residues different from human?) to the human and not expected to behave significantly different. They should emphasize that from 711-777 (in Fig 2 they could put both human and bovine sequences in and show that the numbering is identical and there is only one residue change S747G between human to bovine (note that in Fig 2C, 701 MYH7 should be 711)).

We indeed chose to work on bovine cardiac myosin in order to provide a pseudo-atomic model of IHM, based on the S1-PPS structure that is bovine. In line with this comment and those from reviewer #2 (see answer to his 3rd question), we will release the minimized human model of the IHM of β -cardiac myosin. We have also performed molecular dynamics experiments on the humanized lever arm of the β -cardiac myosin. Bovine and humanized models brought no change in the conclusions of the Converter/ELC interface study (Supp. Fig. 6). Some text has been added in the Mat&Methods section (in the part now called "human β -cardiac myosin", p.19-20) to describe this point: 'Similarly, in order to be sure that the molecular dynamics results are transposable to the human β -cardiac myosin, we humanized the lever arm and performed molecular dynamics during 30 ns (Supp. Fig. 6D). We observed similar dynamics on the human model in the top loop and with the musical chairs' (p.20).

5. This manuscript reports on the post-rigor structure of bovine beta-cardiac myosin at 2.33A resolution and provides the first high-resolution look at the interface between the ELC and the converter domain.

Molecular dynamic simulations of this interface region for WT, as well as for 3 different mutations within the converter domain, provide interesting insights into the potential role of this interface in normal WT function as well as in HCM-disease states. They make an interesting prediction as to the effect of the I736T mutation affecting the IHM state. For the other 3 converter domain mutations, the authors compare their predictions to the biochemical studies of these mutant proteins (Kawana et al in their reference list) but do not mention the fiber studies of these same mutations, which showed force increases of 1.3-1.8 fold rather than decreases in force as seen for the purified proteins. These fiber studies are already referenced in the paper but should be discussed in the context of the molecular dynamics simulations for these mutations.

We agree with the reviewer and we have now added a sentence in discussions to indicate that while the intrinsic force has been slightly reduced by the mutation, the increase in heads linked to the destabilization of the sequestered state would result in more force produced.

Discussions page 16 : “Then, according to our results, even if the R719W and R723G mutations induce a small decrease in force production of individual heads³¹, they also destabilize the sequestered state and increase the number of heads available to produce force, bringing a possible explanation for why fibers from patients carrying these mutations produce more force^{13,14,15}. “ (p. 15-16).

Minor points:

1. Line 192: authors site the Kawana et al paper in reference to the statement “...mutations on fibers mechano-chemical parameters remain modest³¹”. Did they mean to refer to the Kawana et al purified protein studies or rather to refer to fiber studies such as in references 13-15?

Thank you for finding this mistake. This part of the text has been rewritten to make it clearer: “The effects of these HCM causing mutations are different compared to test mutations (R719G and R780E) which resulted in a drastic increase of the structural plasticity of the lever arm. The modest effects of the three HCM mutations on mechano-chemical parameters of S1 fragments³³ confirm that the resulting slight decrease in lever arm compliance keeps the motor functional. The main effect of these three mutations is on the top loop conformation and its dynamics which result in disruption of the dynamic network at the converter/ELC interface.” (p.8).

2. Line 310: ”... (currently limited to a small set of mutations^{31,37,38}” omits references to Sommesse et al (2013) and Bloemink et al (2014), references that are in sup table 2.
The references have been added.

3. Fig 2 legend: Change G471R to G741R.

This has been changed.

4. Fig 3: left panels, cannot see green sphere representing either R719W or R723G

5. Fig. 3: Authors point out that E732 and R783 increase the dynamics at the ELC/converter interface, but it is not clear from the putty diagram in Fig 3 that there are significant fluctuations at the R783 residue, nor do the authors explain what are the different conformations sampled by R783. Initially, we chose R783 to illustrate how the “musical chairs” increase the ELC/converter interface dynamics. R783 indeed oscillates between _{ELC}D130 and _{Plant}E779 in the WT and this dynamic can be modified in some of the mutants. Since E732 is on the top loop and the effects of the different converter mutation on its dynamics are more visual and direct, we finally chose to focus only on this residue to illustrate our concept. The text mentioning R783 has thus been removed.

6. Fig 4: Some of the residue numbers in panel B are incorrect: L300 should be M300, M302 should be L302, D381 should be D382, L386 should be L387.

This has been corrected.

7. Supplementary tables 2 & 3 in their current form are extremely poorly formatted so as to make navigating them very frustrating, if not impossible - all 5 columns should be on the same page! As it is currently laid out, the reader must try to find the page with two columns (including refs) in the second half of the table that correspond to the 3 columns in the first half of the table. Also there should be a key explaining all terms (NPI, etc) as well as the color scheme for the text and the fill. The fill seems to correlate with the pie chart and I understand the coloring of the references based on the experimental system, but what do the colors of the text mean in the first 2 columns?

Thank you for this comment. We took all these improvements into consideration and prepared a new figure 5. In supplementary tables 2 and 3, we removed a column in order to make it more compact and removed some of the colors that were not essential. We kept the color code of the class in order to make a link between Table 1 and Fig. 5. We also detailed the caption to define all the colors and the abbreviations of these tables.

8. 269 “As previously proposed, SOME mutations are predicted to affect the IHM
270 stability since they impact residues found at the interfaces that stabilize the off
271 state^{33,30}”. And
345 “Our results support the previous hypothesis that
346 SOME of the HCM mutations can disrupt the IHM and thus increase the number of
347 heads available to participate in force production¹⁷. This is the case for at least 65%
348 of the set we examined (Fig. 5B).”

References 17,30,33 proposed that MOST HCM mutations are likely to be destabilizing an off-state as a unifying hypothesis, in keeping with what these authors are concluding in this paper.

Line 269, we are referring to our study since, only a part of these mutations destabilizes the IHM. In order to make it clearer, we edited this way “As in the mesa hypothesis, some mutations are predicted to affect the IHM stability since they impact residues found at the interfaces that stabilize the off state^{35,32} (Table 1, Fig. 5A, 5B).” (p.12). We also edited the sentence line 345-346 as following: “Our results support the previous hypothesis that HCM mutations can disrupt the IHM and thus increase the number of heads available to participate in force production¹⁷.” (p.15).

9. Transition between lines 55 and 56 – This seems the best place to bring reference 31 by Kawana et al to attention. Kawana et al used different methods from Brenner to assess the biomechanical properties of beta-cardiac myosin, and furthermore, for the first time, characterized the converter mutations in the context of purified human beta-cardiac myosin, where the HCM mutations occur. In the context of the human beta-cardiac myosin, Kawana et al found that a number of converter mutations resulted in only extremely small changes in ATPase activity, velocity of contraction, and load-dependent force/velocity curves of WT vs mutant myosins. Significant changes in biomechanical properties of the lever arm should have been seen by the loaded in vitro motility assay in particular.

We tried to make this change but it increases the complexity of the text. In our opinion, this information is needed later and we have decided to keep this reference at the end of the results.

10. Line 84 – please insert here which mammalian cardiac myosin was used and how many residue differences there are between it and the human isoform.

This has been corrected as follow: “Here, the cardiac myosin converter/essential light chain interface is described for the first time from a 2.33 Å crystal structure of bovine β -cardiac myosin (98% identity with the human form).” (p.4).

11. 412 “Thus, stabilization of the pre-powerstroke state of the motor is not sufficient to
413 favor the myosin sequestered state and shutting down of motor activity during
414 relaxation. Discovery of drugs that directly affect or favor the formation of the IHM
415 motif would be of exquisite use to control more precisely the motors available.”

This relates to a point made by Anderson et al in a recent BioRxiv publication (Anderson, R.L., Trivedi, D.V., Sarkar, S.S., Henze, M., Ma, W., Gong, H., Rogers, C.S., Wong, F.L., Morck, M.M., Seidman, J.G., Ruppel, K.M., Irving, T.C., Cooke, R., Green, E.M and Spudich, J.A. (2018). Mavacamten stabilizes a folded-back sequestered super-relaxed state of β -cardiac myosin. bioRxiv <http://dx.doi.org/10.1101/266783>) where they propose that mavacamten, now in clinical trials by MyoKardia, causes the myosin S1 head to assume a conformation that favors the folded IHM state. Since Nature Communications allow references to bioRxiv papers, this paper could be referred to. This same paper can be referred to for the authors’ comments regarding the relationship between SRX and IHM on line 71. Anderson et al (bioRxiv <http://dx.doi.org/10.1101/266783>) have presented data with purified proteins confirming that the IHM state is in an SRX state.

Similarly, the Liu et al bioRxiv paper already referred to in this MS can be in the reference list as Liu, C., Kawana, M., Song, D., Ruppel, K.M. and Spudich, J.A. (2018) Controlling load-dependent contractility of the heart at the single molecule level. bioRxiv 258020; doi: <https://doi.org/10.1101/258020>.

We corrected the BiorXiv references. We also added the recent results from Anderson et al., 2018 and that from another paper from Joe Muretta's lab on the topic that has been released on BioRxiv recently (Rohde J.A., Thomas D.A., Muretta J.A. (2018) Mavacamten stabilizes the auto-inhibited state of two-headed cardiac myosin. BioRxiv doi: <https://doi.org/10.1101/287425>). A sentence was added in discussion page 18 and the text is "Discovery of drugs that directly affect or favor the formation of the IHM motif would be of exquisite use to control more precisely the motors available. The inhibitor Mavacamten which has been reported to favor the sequestered state conformation of β -cardiac myosin would be a good candidate for such a treatment^{29,30}".

Reviewers' Comments:

Reviewer #1:

Remarks to the Author:

Authors have addressed all of the questions raised by this reviewer

Reviewer #2:

Remarks to the Author:

The authors have taken great care to respond carefully and fully to the reviewers comments and as a consequence the manuscript is much improved. The image quality in many of the figures are also much easier to read on the page. The work retains the three elements, the new structure of cardiac myosin and the role of the two dynamic converter loops, modelling the interacting head structure giving the best view to date of the interacting surfaces and finally an assessment of the possible role of a range of HCM linked mutations. The three element are now better integrate to a whole document. This is an important paper and will be widely read.

I have only a few minor comments

Minor comments

Line 182/183. " plasticity . The effects " plasticity, the effects

Line 193 " Since on the four" to "Since, of the four "

Line 205 should refs 23 & 32 be 22 & 23?

Line 212/Supplementary Fig 4B. This 2D image of a 3D object gives the impression of a close contact between the free head of Motif 1 and the RLC of blocked head of motif 2. Is this correct and have the authors examined this interface?

Line 307 "close mutations in space" or "mutations close in space" ?

Line 396 A paper published on-line in April addresses the effects of DCM HCM mutations on motor function. "Dilated cardiomyopathy myosin mutants have reduced force-generating capacity."

Ujfalusi et al J Biol Chem doi: 10.1074/jbc.RA118.001938

<http://www.jbc.org/content/early/2018/04/17/jbc.RA118.001938.abstr>

Reviewer #3:

Remarks to the Author:

The authors have answered all of my queries and revised their manuscript answering all the issues I raised. It is an important body of work and should now be published as is.

REVIEWERS' COMMENTS:

Reviewer #1 (Remarks to the Author):

Authors have addressed all of the questions raised by this reviewer

We thank Reviewer #1 for useful remarks that helped us to improve the quality of the manuscript, especially for the figures.

Reviewer #2 (Remarks to the Author):

The authors have taken great care to respond carefully and fully to the reviewers comments and as a consequence the manuscript is much improved. The image quality in many of the figures are also much easier to read on the page. The work retains the three elements, the new structure of cardiac myosin and the role of the two dynamic converter loops, modelling the interacting head structure giving the best view to date of the interacting surfaces and finally an assessment of the possible role of a range of HCM linked mutations. The three elements are now better integrated to a whole document. This is an important paper and will be widely read.

I have only a few minor comments

We thank Reviewer #2 for these good comments. We took particular attention to your comments and edited the manuscript in consequence.

Minor comments

Line 182/183. "plasticity". The effects "plasticity", the effects

This has been corrected.

Line 193 "Since on the four" to "Since, of the four"

This has been corrected.

Line 205 should refs 23 & 32 be 22 & 23?

This is right. This has been corrected.

Line 212/Supplementary Fig 4B. This 2D image of a 3D object gives the impression of a close contact between the free head of Motif 1 and the RLC of blocked head of motif 2. Is this correct and have the authors examined this interface?

This is right that the FH from motif 1 is close to the RLC from motif 2. We chose not to discuss this interface for several reasons (i) it has not been optimized by molecular dynamics, (ii) it involves the RLC which is the less reliable region of our model (obtained by homology modelling), (iii) this interface is quite small and the position of each motif, relative to each other, is obtained from a low resolution map. However, the regions from the FH-motif 1 involved in this interface are also involved in the BH/FH interface (loop 4; a part of the HCM loop and some residues from the U50), so mutations from this region would also be classified as disrupting the IHM in our classification. Future investigation with higher resolution structures of the IHM will probably give more detail on this highly putative interface between motif 1 and 2.

Line 307 "close mutations in space" or "mutations close in space" ?

This has been corrected.

Line 396 A paper published on-line in April addresses the effects of DCM HCM mutations on motor function. "Dilated cardiomyopathy myosin mutants have reduced force-generating capacity."

Ujfalusi et al J Biol Chem doi: 10.1074/jbc.RA118.001938

<http://www.jbc.org/content/>

We added this reference.

Reviewer #3 (Remarks to the Author):

The authors have answered all of my queries and revised their manuscript answering all the issues I raised. It is an important body of work and should now be published as is.

We thank Reviewer #3 for all the useful comments that raised questions allowing to improve the quality of the manuscript.